# The active time model of concurrent choice

J. Mark Cleaveland *

Department of Psychological Science, Vassar College, Poughkeepsie, NY, United States of America

* macleaveland@vassar.edu

## Abstract

The following paper describes a steady-state model of concurrent choice, termed the active time model (ATM). ATM is derived from maximization principles and is characterized by a semi-Markov process. The model proposes that the controlling stimulus in concurrent variable-interval (VI) VI schedules of reinforcement is the time interval since the most recent response, termed here "the active interresponse time" or simply "active time." In the model after a response is generated, it is categorized by a function that relates active times to switch/stay probabilities. In the paper the output of ATM is compared with predictions made by three other models of operant conditioning: melioration, a version of scalar expectancy theory (SET), and momentary maximization. Data sets considered include preferences in multiple-concurrent VI VI schedules, molecular choice patterns, correlations between switching and perseveration, and molar choice proportions. It is shown that ATM can account for all of these data sets, while the other models produce more limited fits. However, rather than argue that ATM is the singular model for concurrent VI VI choice, a consideration of its concept space leads to the conclusion that operant choice is multiply-determined, and that an adaptive viewpoint–one that considers experimental procedures both as selecting mechanisms for animal choice as well as tests of the controlling variables of that choice–is warranted.

## Introduction

One of the core assumptions underpinning models of choice originating from the behaviorist tradition is that animals choose between independent, molar distributions of reinforcement. For instance, the statement that a subject "chose the VI 20-s schedule three times as often as the VI 60-s schedule" would not be judged controversial. This assumption that choice is "of" independent, molar reinforcement schedules is captured best by the well-known matching law [1]. Formally, the matching law has been generalized using Eq 1 [2]

$$\log\left(\frac{B_1}{B_2}\right) = a\log\left(\frac{R_1}{R_2}\right) + \log b \tag{1}$$

where $B_1$ and $B_2$ stand for the frequencies of two behaviors, $R_1$ and $R_2$ stand for the contingent reinforcement frequencies associated with $B_1$ and $B_2$, $b$ represents any bias that the animal might have for one of the two reinforcers, and $a$ represents the sensitivity of the individual to

**Data Availability Statement:** All data is provided in the manuscript or in previously published papers as referenced in the Methods.

**Funding:** The author received no specific funding for this work.

**Competing interests:** The author has declared that no competing interests exist.

the ratio of reinforcement frequencies. In this generalized form, the matching law describes a wide range of molar choice behavior (see [3]). It has been documented with reinforcers that differ in magnitude, with spatial responses, and temporal responses [4–6]. Further, in describing how groups of animals distribute themselves across patches of resources, the matching law also applies, and is known as the "ideal free distribution."

There is no doubt, then, as to the descriptive utility of the matching law. However, its descriptive success has led to the assumption that Eq 1 also provides the outlines of an underpinning mechanism. This jump from the descriptive to the mechanistic can be seen at both the procedural and the theoretical levels. For example, in the operant laboratory the changeover delay, or COD, is a commonly used procedural variable that exists solely to increase the hypothetical control exerted by programmed schedule values [7]. Similarly, at the theoretical level, the assumption that choice is under the control of independent, molar reinforcement distributions lies at the heart of melioration [8, 9], scalar expectancy theory [10, 11], and certain Markov chain models of choice [e.g., 12, 13].

An alternative view of the matching law, however, recognizes that the relation it describes is multiply determined, and therefore, the operative mechanisms underpinning the law will be highly dependent upon the environmental arrangement. More generally, this view of choice behavior, assumes that the mechanisms driving choice are adaptive responses to particular environmental configurations [14–21]. Such a view tends to emphasize the moment-to-moment, or molecular, control of choice, contextual contingencies, and the perceptual constraints of a given organism. The following paper uses this adaptive framework in order to outline a model for choice behavior under concurrent variable-interval (VI) VI schedules of reinforcement. The model is termed the active time model, or ATM. ATM first assumes that the complex temporal environment of typical concurrent VI VI schedules make interval timing difficult, and hence that behavior is under the control of only the most recent interresponse time, termed "active time." Secondly, the model utilizes Markov "states" to define contextual contingencies and a stochastic process in which stays and switches are assumed to be the selected response unit.

The presentation of the model is broken into two sections. In the first section, ATM is briefly described and its core assumptions empirically supported. In the second section four seemingly contradictory data sets are considered. These data sets are matching [1], patterns of interresponse time distributions [22, 23], the correlations of runlengths (i.e., perseveration) to switch probabilities [12, 24, 25], and preferences observed during probes of multiple concurrent VI VI schedules [26–30]. I show that ATM successfully accounts for all four of these data sets, while models such as melioration, a version of scalar expectancy theory, and momentary maximization do not.

## The active time model

ATM is a steady-state, molecular model of concurrent VI VI choice. It suggests that the selection pressures put in place by typical concurrent VI VI experimental contingencies, as well as discrimination constraints related to both procedural variables and the organism in question, encourage subjects to use the time since the most immediate response in order to determine where the next response will occur. In other words, for ATM each response is a temporal marker, and the interval after this marker stochastically controls the next response. In the terminology of the model, the controlling variable is termed the active IRT or "active time."

While engaging with an environment, an organism could potentially track a multitude of time intervals that trigger and reset. In a typical two-choice, two-schedule arrangement there are two types of interresponse times (IRT) that are highly relevant: active time and background

time. As just mentioned, active time corresponds to the time since the most recent schedule choice, while background time refers to the time since the alternative schedule was chosen by the subject. Fig 1 illustrates these two classes of IRT and their significance. Namely, under constant probability VI schedules these two classes of IRT directly determine reinforcement probabilities via Eq 2 (or approximations [31])

$$P_i = 1 - e^{-t_i/\lambda_i} \tag{2}$$

where $P_i$, the probability of reinforcement at Choice $i$ depends upon the time, $t_i$, since last choosing Choice $i$ and the average reinforcement rate, $\lambda_i$, assigned to Choice $i$. In order to apply Eq 2 to concurrent choice experiments, one simply iterates Eq 2 for each choice. Fig 1, for instance, shows how the probabilities of reinforcement change when an animal responds solely at either of two alternatives in a concurrent VI 20-s VI 60-s schedule.

For simplicity, Fig 1 assumes a response every 6 s as in, say, a discrete-trial procedure. As can be seen, at very short intervals active IRTs correspond to low probabilities of reinforcement. In contrast, short active time intervals will be associated with relatively higher probabilities of reinforcement at the background schedule. As both active and background IRTs increase, the relative value of the associated reinforcement probabilities will depend on the specific schedule values in place. However, regardless of schedule values, short active times always correspond to lower reinforcement probabilities than those to be had at the background schedule. Also, if the active schedule is relatively rich, then its reinforcement probabilities will always grow at a faster rate than those at the background schedule. In such cases, an optimal choice strategy, when responding at the relatively rich VI schedule, would be for the subject to show a bias for switching at short active IRTs while staying after longer active IRTs.

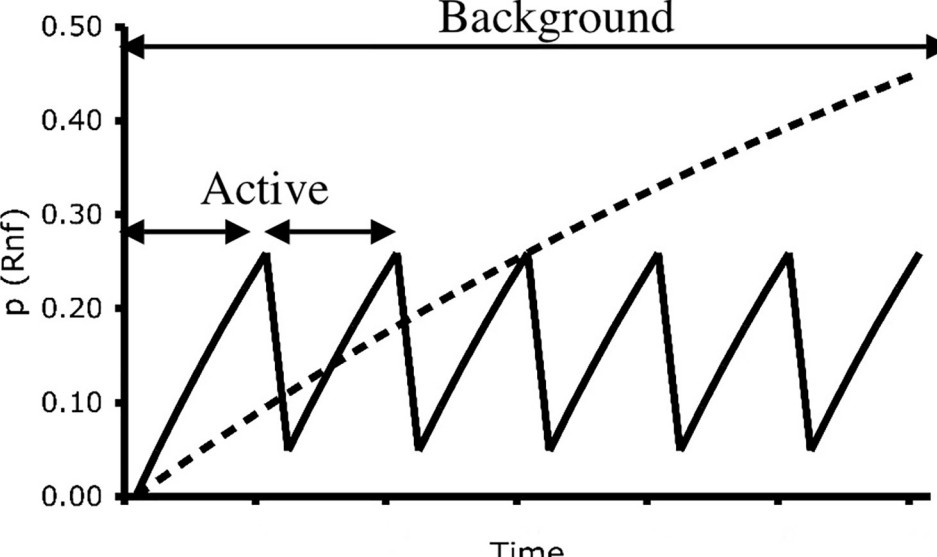

**Fig 1. Active and background interresponse times.** The figure assumes that a subject responds at a fixed interval, and shows the relationship between interresponse times (IRTs) and reinforcement probabilities during concurrent variable-interval (VI) VI schedules of reinforcement. In this case, the subject responds solely on the richer of two concurrent schedules of reinforcement. As it does so the time since the last peck to the alternative, or "background IRT" increases, leading to a steady increase in the probability of reinforcement at the background schedule. Each choice of the active schedule resets its IRT—the "active IRT"—which causes the probability of reinforcement at the active schedule to reset to zero.

This relationship between when an animal responds and where it responds is at the core of ATM. In skeletal form, ATM proposes that an IRT generator determines when the next response will occur. Where this response will occur is then determined by an active time function that relates active time to switch-stay probabilities. This interaction between an IRT generator (when) and an active time function (where) is formalized in ATM by a semi-Markov process combined with a renewal process [32].

Markov processes fall into a variety of categories, of which a semi-Markov process is a somewhat complex member. However, all Markov processes rely on the fact that in the described system behavior is controlled by a discrete set of environmental variables, or states, and that the probability of moving from any given state to another depends solely on the resident state. Mathematically the Markov property is defined by

$$P(X_n|X_0, \ldots, X_{n-1}) = P(X_n|X_{n-1}) \tag{3}$$

where $P(X_n)$ gives the probability of being in state $X_n$. In words, the Markov property states that the probability of being in state $X_n$, given past residence in any number of the preceding states, depends only on the most recent state, $X_{n-1}$. If there were two states, $i$ and $j$, then we could say that $P_{ij} = (X_n = j \mid X_{n-1} = i)$. The Markov property, then, allows us to create a transition matrix that perfectly describes the

behavior within a system. In a two-state system with states 1 and 2 our matrix would be as follows

$$\begin{array}{cc} & \begin{array}{cc} 1 & 2 \end{array} \\ \begin{array}{c} 1 \\ 2 \end{array} & \begin{pmatrix} 1-p & p \\ q & 1-q \end{pmatrix} \end{array} \tag{4}$$

where $p$ is the probability of moving from state 1 to state 2, and $q$ is the probability of moving from state 2 to state 1.

If the matrix provided in Eq 4 were sampled at regular, discrete time intervals, then we would term the model a *Markov chain*. For instance, if a pigeon responded once per second, and switched between two choices with probabilities set by the matching law, then a Markov chain could be used to describe the bird's behavior. Pigeons, though, do not normally respond at discrete, regular time intervals, and in fact many systems are best described in continuous time. A Markov model describing a continuous time system is referred to as a *Markov process*. Unlike a Markov chain, the transition probabilities in a Markov process describe the probability of moving from one state to another in some arbitrarily fixed, interval of time, $\Delta t$. That is, the response unit in a Markov process is not so much discrete choices but rather allocated time intervals. If we let $N(t)$ be the process that generates these time intervals, then our transition matrix for a two-state system would be

$$\begin{array}{cc} & \begin{array}{cc} 1 & 2 \end{array} \\ \begin{array}{c} 1 \\ 2 \end{array} & \begin{pmatrix} N(t)(1-p) & N(t)p \\ N(t)q & N(t)(1-q) \end{pmatrix} \end{array} \tag{5}$$

In words the matrix states that when in State 1, a switch to State 2 occurs with a probability $p$ at each arrival of $N(t)$. Conversely, when in State 2, a switch to State 1 occurs with a probability $q$ at each arrival of $N(t)$.

Since ATM describes behavior in continuous time it is related to a Markov process, and $N(t)$ is considered to be the process that generates active IRTs. In ATM, though, switch probabilities are not fixed as shown in Eq 5. Rather, ATM assumes that while the organism resides in a particular state, the probability of switching changes as a function of $N(t)$, which is equivalent to the time since the most recent decision of whether to stay or switch. Such a process is termed a *semi-Markov process* [32]. A semi-Markov process is a generalization of the simpler Markov process in which the assumption of constant transition probabilities, e.g., p and q from above, has been removed. These points can be summarized as follows. Again, let $N(t)$ be the process that generates time intervals, and $A(t) = A_{N(t)}$ be the state-specific function that describes how the probability of switching from that state to another changes as a function of $N(t)$. For ATM $A(t)$ is termed the active time function, and our transition matrix for a two-state system would be

$$
\begin{array}{cc}
& \begin{array}{cc} 1 & \qquad\qquad 2 \end{array} \\
\begin{array}{c} 1 \\ \\ 2 \end{array} &
\left(
\begin{array}{cc}
N(t)(1 - A_{1N(t)}) & N(t)A_{1N(t)} \\
\\
N(t)A_{2N(t)} & N(t)(1 - A_{2N(t)})
\end{array}
\right)
\end{array}
\qquad (6)
$$

The matrix in Eq 6 states that active IRTs are generated according to the state-independent function $N(t)$. $A(t)$, though, is state specific, so that the probability of switching from State 1 to State 2 is given by $A_{1N(t)}$, while the probability of switching from State 2 to State 1 is given by $A_{2N(t)}$.

Fig 2 summarizes Eq 6 in graphical format. To reiterate, ATM makes the following assumptions:

1. <u>State space.</u> it assumes that an animal, at any given moment, is in a particular choice state defined by experimental contingencies.

2. <u>Active time control.</u> Each state is associated with a state-specific active time function that determines the probability of switching on a moment-to-moment basis, $\alpha$ and $\beta$ in the figure. Response units, then, are "stays" and "switches."

3. <u>IRT generator.</u> These moments, though, are themselves determined by a single, independent IRT generator, $N(t)$.

In summary, ATM assumes that an IRT generator will determine *when* the next response will occur, and that a state-specific active time function will determine *where* this response occurs. In a two-state environment, though, response units merely consist of "stay" and "switch."

## Empirical support of core assumptions

**Molecular control by active time.** ATM's core assumption is that active time is a primary controlling variable of concurrent VI VI choice. In support of this assumption, Figs 3 and 4 provide active time functions drawn from discrete-trial, free-operant and multiple concurrent procedures [24, 27, 29]. Fig 3 provides functions drawn from the relatively rich VI of a pair, while Fig 4 provides functions drawn from the relatively lean VI. In every case subjects were most likely to switch at short active times. In Fig 3 the predominant pattern is that as active time increases, the probability of switching to the relatively lean schedule decreases. This fits the hypothesis that subjects use active time as a "rule of thumb" for reinforcement probabilities. Fig 4 provides a more complicated picture. In general, it shows that as active time

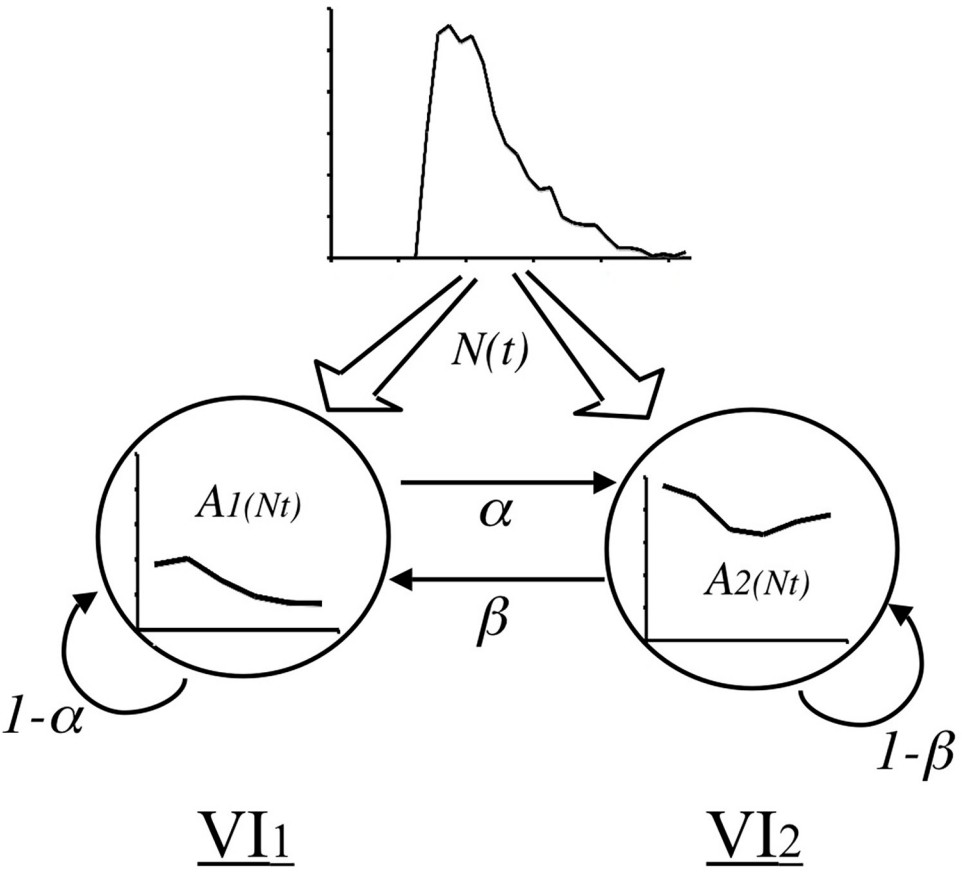

**Fig 2. A graphical representation of the active time model.** ATM assumes that an independent interresponse time generator N(t) determines when a subject will respond. The generated interval is then used in a semi-Markov process in order to determine where the subject will respond. This later determination depends on the resident state of the subject and crucially on a state-specific function, A(t) that relates a sampled interval to switch probabilities. The resultant probability, a or b in this case, is used to determine the choice of the subject.

increases at the lean schedule, the likelihood of switching remains high. However, in some cases, the likelihood of switching out of the lean VI also decreases at longer active times. Regardless, every subject in each of the three procedures shows correlations between active IRTs and switch probabilities that are suggestive of active time control.

That IRTs can control behavior, i.e., serve as discriminative stimuli, has been well established in other studies, as well. Shimp [33, 34], for example, differentially reinforced pigeons for patterns of pecking that consisted of short and long IRTs. He then inserted matching-to-sample probes in which pecks to a red key were reinforced after short IRTs, whereas after longer IRTs pecks to a green key were reinforced. Shimp found that pigeons were able to learn and perform the matching-to-sample task at over 90% accuracy. Even after an 8 s. delay was inserted between the IRT and the matching-to-sample probe, accuracy remained greater than chance. Similarly, visual stimuli trained with similar rates of responding (i.e., short or long IRTs) have been found to substitute for one another in matching-to-sample tasks, which suggests that IRTs, rather than the visual stimulus, is the relevant controlling stimuli in such circumstances [35–37].

Studies have also found that IRT frequencies are responsive to reinforcement contingencies [38–42]. Tano & Sakagami [42], for example, subjected rats to reinforcement contingencies in

# Rich Active Time Functions

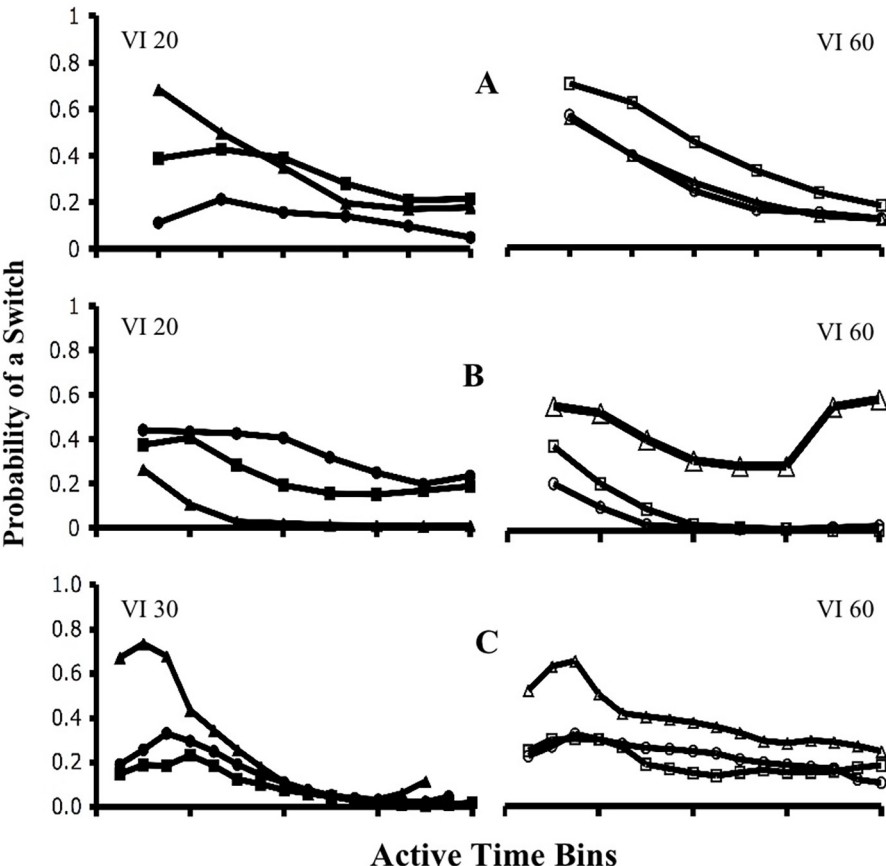

**Fig 3. Rich schedule active time functions.** This figure shows the active time functions for the richer of six different concurrent VI VI pairs drawn from three different procedures. Solid and un-filled labels indicate a VI pair whose leaner partner is shown in Fig 4. A) Functions for six different pigeons experiencing a discrete-trial procedure [24]. On the left subjects experienced a VI 20-s VI 60-s schedule, while on the right subjects experienced a VI 60-s VI 180-s schedule. The time bins were < 4, 4–5, 5–6, 6–7, 7–8, and >8 seconds. B) Functions for three subjects trained first in a VI 60-s VI 180-s schedule and then in a VI 20-s VI 60-s schedule [24]. The time bins are in 5-s intervals up to 4 s. C) Functions for three birds trained with multiple concurrent, free-operant VI VI schedules and a Findley procedure [29]. The schedules utilized were a VI 30-s VI 60-s and a VI 60-s VI 120-s, and the time bins are in 25-s intervals up to 4 s.

which molar feedback functions were flat and, therefore, similar to a VI schedule. Individual IRTs, though, were reinforced in a manner similar to a variable ratio schedule of reinforcement. In every case the rats emitted IRTs that were similar to those found under variable ratio schedules. Similarly, it has long been known that the differential reinforcement of IRTs can produce molar rates that approximate those of a VI schedule [40, 41].

That IRTs can control choice and be selected by reinforcement, therefore, is not controversial. However, ATM does make the unique claim that of the two types of IRT extant in a typical two-choice experiment, only the active IRT controls choice. In other words, ATM assumes that there are limitations in a subject's ability to discriminate multiple, overlapping IRTs during concurrent VI VI experiments. The evidence for such a limitation is, in fact, mixed. Some experiments have found that subjects are capable of discriminating between multiple, simultaneously occurring temporal intervals. For instance, in a situation in which two intervals start

# Lean Active Time Functions

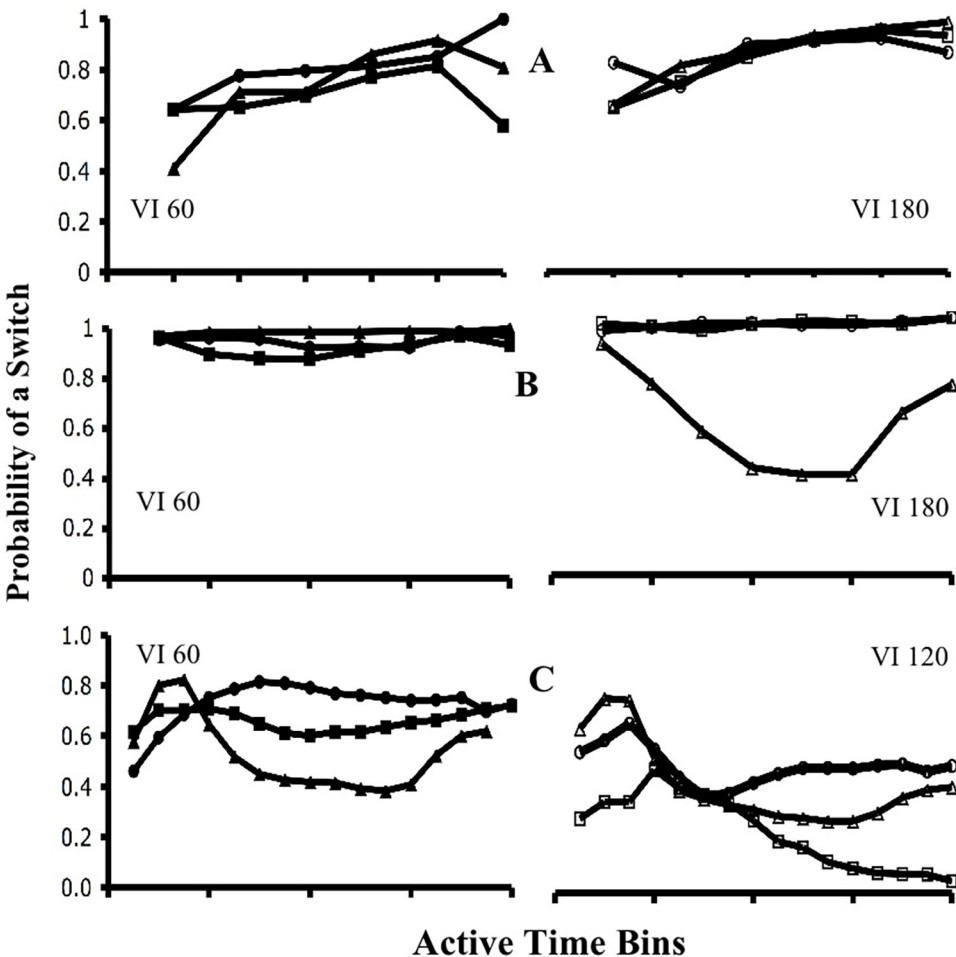

**Fig 4. Lean schedule active time functions.** This figure shows the active time functions for the leaner of six different concurrent VI VI pairs drawn from three different procedures. Solid and un-filled labels indicate a VI pair whose richer alternative is shown in Fig 3. Relevant time bins are given in Fig 3. A) Functions for six different subjects experiencing a discrete-trial procedure [24]. B) Functions for three subjects trained under a standard free-operant procedure [24]. C) Functions for three birds trained with multiple concurrent, free-operant VI VI schedules and a Findley procedure [29].

in parallel, both rats [43] and humans [44, 45] produce behavior indicating that separate end-points can be discriminated.

A concurrent VI VI procedure presents a much more complex interval timing structure, though. Background and active IRTs do not start in parallel, although they do on occasion end in parallel, and while the background IRT is accumulating, the active IRT might start and end several times. Also, the stimulus trigger initiating each IRT is not an external event, but a behavior that is common to both intervals (i.e., a key peck). Finally, the multiple IRTs in a concurrent VI VI schedule occur within the same sensory modality. Given this complexity, one would expect temporal discriminations to be quite difficult during concurrent VI VI

contingencies. Indeed, and in contrast with the active time functions shown in Figs 3 and 4, to date there is no evidence that background IRTs control choice [12, 24, 25, 46].

**Stay/switch response units.** ATM not only claims that active time is a primary controlling variable in concurrent VI VI schedules of reinforcement, it also assumes that the response units that this stimulus controls are stays and switches. In other words, ATM assumes that the frequency-dependent nature of concurrent VI VI schedules selects both the relative frequency of switches and stays as well as when they occur. In contrast, some other popular accounts of concurrent VI VI schedules assume that the primary response units are mutually exclusive rates of responding to two or more schedules, and that the overall or local rates of reinforcement from these schedules differentially select among these units [e.g., 8, 12, 47]. Whether these differences are due to procedural variables will be considered later in the paper. Nonetheless, ATM's assumptions concerning response units dovetail with MacDonall's stay/switch model [48–50] and by work done by others [51–53].

MacDonall's stay/switch model is an increasingly supported molar model of choice that conceives of each alternative as a state in which the subject either stays or switches. As such, the model views the traditional concurrent VI VI schedule as consisting of four reinforcement schedules. For example, in a VI 40-s VI 80-s schedule stay$_{VI40}$ is reinforced with a VI 40-s schedule and switch$_{VI40}$ (i.e., switches out of the VI 40-s schedule) is reinforced with a VI 80-s schedule. Conversely, stay$_{VI80}$ and switch$_{VI80}$ are reinforced with a VI 80-s and a VI 40-s schedule, respectively.

For standard concurrent VI VI experiments, switch-stay contingencies will always be symmetrical. However, they need not be. One could, for example, arrange an alternative in which stays were reinforced with a VI 40-s schedule, and switches were arranged with a VI 40-s schedule, with a similar arrangement holding for the "VI 80" alternative. The question is whether the generalized matching law (Eq 1), with an assumption of singular, schedule-directed response units, can fit such an asymmetrical arrangement. The answer is, no. In contrast, the stay/switch model fits both the symmetrical and asymmetrical arrangement of concurrent VI VI schedules of reinforcement [49, 50].

That stays and switches are the primary response units of concurrent VI VI schedules is further supported by work examining "hold" functions [52]. Concurrent VI VI schedules are often programmed using a pre-determined set of intervals [e.g., 31]. If the subject is responding at one choice, and a reinforcer is determined to occur at the alternative, then that reinforcer is held until the subject claims it. This would correspond to a "hold function" of 1, meaning that if a reinforcer is set up, it is always held until claimed. Jensen & Neuringer systematically altered the hold function–in essence altering the relative reinforcement probabilities of switch and stay responses–and this led to changes in choice preference that correlated with the relative probability that a switch would be reinforced. When hold functions approached 0, choice tended to become exclusive to the richer VI schedule. Importantly, low hold functions may be thought of as making concurrent reinforcer distributions more independent of one another. This is, after all, the logic used to justify a changeover delay (COD). Jensen & Neuringer's approach, though, emphasizes the discrimination of stays and switches rather than responses made to programmed reinforcer distributions, as is assumed by the traditional approach to matching (Eq 1).

Finally, the relevance of switch and stay response units is shown in work by Machado [53] using frequency-dependent schedules of reinforcement. Frequency-dependent schedules, of which concurrent VI VI schedules are a subset, are characterized by the differential reinforcement of response units based on their relative frequency [54–56]. What Machado found was that when response units were defined in terms of a single stay or switch, pigeons quickly learned to maximize reinforcement by alternating L (left) and R (right) pecks. Further, when

response units were defined in terms of two responses—LL, LR, RL and RR–some of the pigeons learned to maximize reinforcement by emitting each pair once, in sequence (e.g., RRLLRLLR). When the response unit was defined by triplets, though, pigeons failed to learn the optimal sequence. Instead, they emitted choices probabilistically. A similar result has been found with budgerigars in which the response class consisted of call types [57].

Machado's data suggests two things. First, switches and stays can be differentially reinforced. However, if subjects are unable to discriminate the relative rates of reinforcement for different patterns of stays and switches, then they will come to emit stays and switches stochastically. ATM adds to Machado's account by suggesting that it is not just stays and switches that are reinforced but rather stays and switches at particular times.

**Context dependency and state space.** One final assumption of ATM concerns the use of a Markov framework and its assumptions of stochastic movement within a "state space." Markov chains have been shown to be extremely useful for modeling a large number of physical, biological, and behavioral phenomena [32, 58, 59]. Further, there is reason to believe that Markov models can be applied productively to concurrent VI VI choice [12, 13, 24]. At this point, then, the question is no longer whether concurrent VI VI choice is stochastic, but whether the use of a state space is necessary. Isn't a state space simply equivalent to the number of variable-intervals arranged in the environment?

The answer to this question is no. State space is largely dependent on the experimental contingencies put into place by the experimenter. Fig 5 provides a state-space representation of three common concurrent VI VI procedures. In the top diagram two states are given, each corresponding to a choice of either of two VI schedules. Cleaveland [24] found that such a two-state representation was adequate to describe a discrete-trial experiment in which pigeons were required to wait an interval of time between each choice. However, when birds were allowed to respond freely (i.e., a free-operant procedure) Cleaveland found that additional states were required in order to describe their behavior. He hypothesized that in the free-operant procedure the first peck to a schedule is qualitatively different from all subsequent pecks to the schedule. This finding relates to Machado's work [53] described above, in that the free-operant procedure seemed to allow birds to discriminate between pairs of responses (e.g., RL, LL, LR, RR), while the discrete-trial procedure appeared to limit discrimination to single responses. Finally, the lowest graphic in Fig 5 provides a hypothetical representation for a Findley procedure [60]. In such a procedure, schedule stimuli appear on a single response key and a second key (CO) allows the subject to switch back and forth between the schedule stimuli. Using such a procedure, Williams and Bell [61] found evidence that choice was under the control of overall rates of reinforcement. In contrast, using a free-operant procedure, Williams and Bell [30] found evidence suggesting that choice was determined by learned patterns of switching.

A state space, then, reflects the controlling variables of an experimental procedure rather than simply the variable-interval schedules, themselves. For ATM state spaces are defined by active time functions and the discriminative stimuli that define the location of a particular choice. However, it is very probable that given the appropriate stimulus environment, concurrent VI VI choice might be under the control of local reinforcement rates [8, 62], molar rates of reinforcement [62, 63], reinforcement probabilities [54, 55], interreinforcement intervals [64], response-reinforcement associations [65], reinforcement traces [5], or active times [24]. In other words, a procedural manipulation designed to test between two models, might not be so much a "test of" as a "selection of" the models. The task, then, is not to specify *the* mechanism underpinning choice, but to delineate the relationship between environmental regularities and controlling variables–the state space—that describe behavior. A Markov framework with its conception of states, in other words, is an essential element not just of ATM, but of models of concurrent VI VI choice in general.

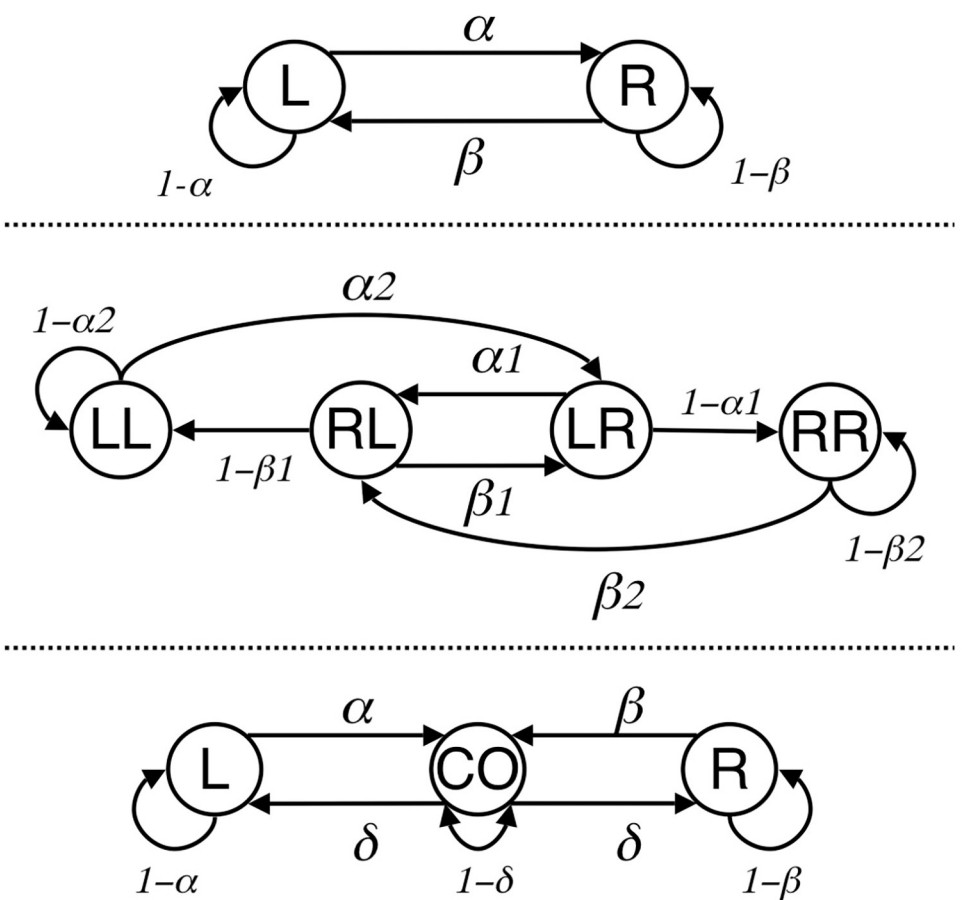

**Fig 5. Markov state representations for three types of concurrent VI VI procedures.** A)Shows a first-order Markov process which Cleaveland [24] found to fit a discrete-trial procedure. Here, the only discriminative stimulus consists of a single schedule choice, e.g., L or R. B) Shows how a second-order Markov process represents the sequential dependencies often found with free-operant procedures. C) A Markov representation for a Findley procedure [60]. Here the subject makes a choice via a single changeover key (CO) that controls the appearance of alternatives on a second, schedule key. In some cases d, the probability of selecting a reinforcement schedule (R or L) is set to 1 [e.g., 30] which forces the subject to emit the response pattern CO-L or CO-R. In other cases d is not determined by the experimenter but by the subject [e.g., 27].

## Methodology

The first half of this paper described the active time model (ATM). It also discussed the assumptions underlying ATM. These assumptions concern IRT stimulus control, stay-switch response units, and a Markov framework that emphasizes the selective pressures exerted by procedural variables. Empirical support was offered for each of these underlying assumptions. The remainder of the paper compares the predictive results of ATM with three other models of operant choice: melioration [8, 9], a version of scalar expectancy, termed SET* in this paper [28], and momentary maximizing [22, 23, 54, 55].

### General parameters

The simulations of all four models were programmed using the Xojo programming language. Brief outlines of each model and the utilized, specific parameters are given below. All simulations, however, were conducted with the following general assumptions and parameters:

1. All simulations were of two-choice procedures commonly used in operant behavior studies.

2. Interresponse times (IRTs) were generated prior to a model's decision rule being applied, and the same IRT distributions were used for all four models.

3. For free-operant simulations response rates were programmed using an exponential function of form -ln(n)/λ + .2, where n is a random number drawn from the range 0 to 1 and λ is the desired average response rate. Unless otherwise indicated in the results, a response rate of 1/s was used for all simulations.

4. For discrete-trial simulations response rates were programmed using a Gaussian distribution with a mean of 6 s and a variance of 1. The distribution was truncated such that the lowest value was 2 s and the highest value was 10 s.

5. The time to switch from one alternative to another alternative was set at 0.3 s. This time was in addition to whatever IRT had been fed into the decision rule (i.e., "switches" cost an additional 0.3 s).

6. Rewards were programmed to occupy 2 s of time.

7. Two separate clocks programmed reinforcer deliveries. When a reinforcer occurred at one alternative, the clock for that schedule was reset and did not resume until the reinforcer period was finished. However, the schedule clock at the non-reinforced alternative continued to run.

8. No changeover delay (COD) was simulated.

9. Simulations ran for 20,000 responses. The last 2,000 responses were saved to data files in order to generate the clock spaces shown in the results (Figs 10 & 11).

## Melioration-specific parameters

Melioration hypothesizes that the choice behavior of pigeons under concurrent VI VI schedules of reinforcement is controlled by differences in local rates of reinforcement [8, 9]. Melioration is derived directly from the matching law by noting that both behavioral frequencies ($B_i$) among a set of alternatives and time allocation ($T_i$) to these alternatives tend to match contingent reinforcement frequencies ($R_i$). That is for two alternatives,

$$\frac{B_1}{B_2} = \frac{T_1}{T_2} = \frac{R_1}{R_2} \tag{7}$$

From Eq 7 it follows that matching holds when

$$\frac{R_1}{T_1} = \frac{R_2}{T_2} \tag{8}$$

or, in other words, when the rate of reinforcement per time spent at Alternative 1 equals the rate of reinforcement per time spent at Alternative 2. By simply always choosing the alternative with the higher local rate of reinforcement, an organism will tend to produce matching at the molar level. It should be noted that melioration is silent as to the time window over which local reinforcement rates are calculated. Therefore, in the simulations of this model windows of 15, 30 and 60 time units were utilized. In addition, the following two assumptions were made:

1. Local rates of reinforcement were calculated by creating an array of the last $n$ (15, 30, 60) instances of each of three variables: the VI choice, the IRT that had been generated just

prior to the VI choice, and whether that choice resulted in a reinforcer. Local rates of reinforcement were then calculated by dividing the total number of reinforcers assigned to each choice within the array by the sum of the IRTs assigned to each choice within in the array.

2. When local rates of reinforcement were equal to one another or both equal to zero, the simulated organism chose either alternative with a probability of 0.5

## SET* specific parameters

Scalar expectancy theory (SET) emerged as a description of interval timing [10, 11], and its use has been much extended since [64]. With regards to concurrent VI VI behavior, SET assumes that concurrent VI VI choice is controlled by an expectation of time until food. Accordingly, as an animal experiences concurrent VI VI schedules, SET hypothesizes that two independent distributions of remembered interreinforcement intervals are created. At the moment of choice, the animal samples its memory distributions, compares these samples, and chooses the stimulus associated with the smaller interval.

Gibbon [28], proposed a variant of SET (termed SET* in this paper) that extends this framework in an attempt to account for a broader range of data. SET* adds two features to SET. First, it assumes that interreinforcement memory distributions, rather than directly determining choice, do so indirectly by establishing the overall switch probabilities associated with a particular stimulus. Secondly, SET* assumes that these switch probabilities are sampled at a rate that is proportional to the overall reinforcement rate for a pair of concurrent schedules. In making predictions for SET*, therefore, the following assumptions were made

1. For Data Sets 1–3, SET* corresponds to a Markov process in which switch probabilities are set by the ratio of molar reinforcement rates.

2. Contra Gibbon [28] it was assumed that each time interval generated by the IRT distribution corresponded to an overt response either staying or switching.

3. For Data Set 4, switch probabilities were multiplied by the ratio of overall reinforcement rates from each trained pair of concurrent VI VI schedules in order to makes predictions of the novel choice pairings described in this data set.

## Momentary maximizing specific parameters

Momentary maximization [22, 23, 54, 55] hypothesizes that concurrent VI VI choice is controlled by a comparison between reinforcement probabilities at each of the available alternatives at the moment of choice. As mentioned earlier, reinforcement probabilities in concurrent VI VI schedules are usually given by Eq 2. This leads to concurrent changes in reinforcement probability as illustrated in Fig 1. In other words, momentary maximizing in reality assumes that subjects are sensitive to changes in both the active and background IRTs. An animal following a momentary maximizing strategy simply selects the choice associated with the IRT that has the highest momentary probability of reinforcement. Simulations of momentary maximization required only two assumptions:

1. After an IRT, reinforcement probabilities were calculated according to Eq 2.

2. To calculate the prospective reinforcement probabilities of choosing an alternative that required a switch, the minimum switch time of 0.3 s was added to that choice's IRT. This addition was "hypothetical" in the sense that if the simulated organism chose to stay at the

current alternative, this switch time was not added to the accumulating background interresponse time.

### ATM specific parameters

To simulate the results of the active time model, an average organism was created from past experimental findings. Thus,

1. For the free-operant simulations used in Data Sets 1–3, active time switch functions were created by averaging across the five subjects used in [24]. The average was only taken from the VI 20-sVI 60-s schedule and involved 0.5-s bins up to 4.0+ s. The resultant function for the rich VI was: .43, .36, .19, .13, .08, .08, .18, and .18. The function for the lean VI was: .90, .90, .80, .75, .73, .72, .80, and .88.

2. For discrete-trial simulations used in Data Sets 1–3, active time switch functions were created by averaging across the six subjects used in [24]. Time bins (s) were $< 4$, 4–5, 5–6, 6–7, 7–8, and $> 8$. The resultant function for the rich VI was: .5, .42, .31, .21, and .14. The function for the lean VI was: .63, .73, .79, .85, .89, and .85.

3. For Data Set 4 the predictions shown in Fig 14 were generated by using the average active time functions listed above, as well as that for the VI 60-s schedule in the VI 60-s VI 180-s pair in [24]. The VI $60_{180}$ function was: .32, .29, .25, .18, .14, .13, .13, and .13.

### Data sets

Four data sets were selected over which to compare the simulated results of the four models. These data sets are

Data Set #1: Simple two-choice matching [1],

Data Set #2: Patterns of interresponse time distributions [22, 23],

Data Set #3: Correlations between runlengths (i.e., perseveration) and switch probabilities [12, 24, 25, 46]

Data Set #4: Preferences observed during probes of multiple concurrent VI VI schedules ([26–29].

In the results, each of these data sets is described along with the outputs generated by the four models.

### Results

### Data set 1: Matching

In its most basic form, matching is the finding that animals tend to arrange their choices in proportion to the relative amount of reinforcement each alternative delivers [1]. This molar relationship describes a wide range of molar choice behavior from the laboratory [e.g., 3]. Further, Houston [66] showed that the generalized matching law correctly fits the foraging allocations of pied wagtails in the wild. Finally, one can also show mathematically that in concurrent VI VI schedules, matching produces the choice distribution that maximizes the overall rate of reinforcement. Given its success, for a concurrent VI VI choice mechanism to be acceptable, it must yield matching at the molar level.

Since melioration is directly derived from the matching law, it follows that it should produce molar matching. Table 1 shows that simulations of melioration across two concurrent schedules do indeed produce choice proportions that approximate the scheduled reinforcement ratios. However, Table 1 also shows that the degree to which melioration produces

matching depends to a large extent on the time window over which local rates of reinforcement are calculated. Keeping VI values constant, larger time windows will produce response proportions that are more in line with the matching law, while shorter time windows trend towards indifference (so-called "undermatching"). This effect of window size can be understood intuitively. With a window of 1, local "rates" of reward are almost always 0, and choice behavior would presumably be random (this would depend upon the specific implementation of melioration). This would produce indifference between the two choices. As the time window for calculating local rates grows, a more accurate approximation of the actual, programmed schedules becomes possible. Thus, with a time window equal to the duration of a session, the local rates would be equivalent to the programmed reinforcement rates divided by the proportion of allocated time. For example, in the case of a concurrent VI 20-s VI 60-s schedule reinforcement rates are 3/min and 1/min., respectively, which a large window would accurately represent. Spending ¾ of the time at the VI 20-s schedule and ¼ of the time at the VI 60-s schedule would produce equal local rates of 4/t. at each alternative. This, of course, is the matching law.

As with melioration SET* is able to formally derive the matching law [64]. Given that VI schedules are typically programmed according to a Poisson process, the experienced and remembered interreinforcement intervals, despite the scalar property of variance, will conform to exponential distributions. Imagine, therefore, two concurrent VI schedules $VI_1$ and $VI_2$. As the animal experiences these two schedules it will establish two independent memory distributions with rates given by $\lambda_1$ and $\lambda_2$. The relative choice frequencies are given by the probability that a memory sample, $S$, drawn from $\lambda_1$ will be shorter than one drawn from $\lambda_2$, or by

$$P(S_1 > S_2) = \frac{\lambda_1}{\lambda_1 + \lambda_2} \tag{9}$$

What Eq 9 shows is that the probability of choosing $VI_i$ is given by the relative, remembered rates of reinforcement. Since the remembered rates of reinforcement are proportional to the actual, programmed rates of reinforcement, Eq 9 recapitulates the matching law. The simulation of this result is trivial, namely because it simply requires the flipping of a coin weighted by Eq 9. This will, by definition, produce perfect matching as is evident in Table 1.

Momentary maximizing, too, can be formally derived from the matching law. Eq 7, in addition to supporting melioration, also implies that when matching holds, reinforcement per unit of behavior at Alternative 1 equals the reinforcement per unit of behavior at Alternative 2.

**Table 1. Simulations of matching for four models of VI-VI choice.**

| Model | | Free Operant | | Discrete-Trial | |
|---|---|---|---|---|---|
| | | Proportion rich | | Proportion rich | |
| | ** | VI20-VI60 | VI60-VI180 | VI20-VI60 | VI60-VI180 |
| Melioration | 15 | 0.62 | 0.55 | 0.62 | 0.61 |
| | 30 | 0.63 | 0.58 | 0.60 | 0.62 |
| | 60 | 0.69 | 0.61 | 0.63 | 0.62 |
| Momentary Max | | 0.67 | 0.68 | 0.71 | 0.71 |
| SET* | | 0.75 | 0.74 | 0.75 | 0.75 |
| Active Time | 1/s | 0.72 | 0.63 | 0.75 | 0.75 |
| | 2/s | 0.68 | 0.72 | 0.75 | 0.75 |

Provides proportion of rich schedule to lean schedule choice for model simulations. For melioration three windows were utilized over which local rates were calculated. For ATM two response rates were simulated.

This is equivalent to stating that when matching holds, the probability of reinforcement at Alternative 1 equals the probability of reinforcement at Alternative 2. Therefore, by always choosing the alternative with the momentarily higher probability of reinforcement, an organism should tend to oscillate around matching at the molar level. In fact, Table 1 shows that for the simulations considered here, momentary maximizing produced choice distributions that were consistently below matching (i.e., undermatching). This is, in fact, an artifact. The simulation used for this paper assumed a .3-s changeover interval *after* a decision was made to switch. In other words, reinforcement probabilities were not perfectly tracked. Undermatching is, however, a more robust finding than matching, at least for pigeons behaving under concurrent VI VI schedules of reward [3, 67].

Unlike melioration, SET* and momentary maximizing, whose predictions of matching may be formally derived from a priori assumptions, ATM's predictions strictly depend on the active time functions measured from the actual behavior of a subject. As yet, there is no formal derivation of these functions from reinforcement principles. However, as Table 1 shows, for the simulations considered here, ATM consistently yielded predictions that approximated the matching law as well as melioration and momentary maximizing. Like the latter two mechanisms, ATM tended to produce undermatching. Further, as Table 1 indicates, the degree of undermatching depended on the response rate used. A rate of 1 choice /s. produced choice proportions closer to matching than a rate of 2 choices /s. The reason for this is that the difference between the rich and lean functions is greater at 1/s. than 2/s for the active time functions used in these simulations (see Methods).

A strong prediction of ATM, then, is that changes in response rate will produce momentary changes in molar choice proportions as a function of the existing, underlying active time functions. In fact, an experiment by Misak & Cleaveland [68] produced just such a result. Fig 6 provides a schematic of the experimental logic as well as its essential data.

In their experiment Misak & Cleaveland trained pigeons on a discrete-trial, multiple concurrent VI VI procedure with a VI 20-s VI 40-s and a VI 40-s VI 80s. The logic of the experiment was that, although the two VI 40-s schedules were objectively identical, the underlying active time functions trained to their paired stimuli would be quite different. The VI 40-s stimulus paired with the VI 20-s schedule would elicit a "lean" active time function, while the opposite would be the case for the VI 40-s stimulus paired with the VI 80-s schedule (Fig 6A). During a series of probes, the pigeons were presented with the two stimuli associated with the VI 40-s schedules and either forced to respond at relatively long or relatively short intervals (Fig 6B). As ATM predicts choice proportions during the "short probes" were close to indifference, while choice proportions during the "long probes" showed extreme preference for the VI 40-s stimulus that had been trained in the VI 40-s VI 80-s schedule (Fig 6C). Such a result provides strong experimental evidence for ATM's assertion that choice proportions obtained during concurrent VI VI schedules—matching, undermatching and overmatching–are a function of underlying active time switch functions.

## Data set 2: Molecular structure of choice

All of the choice mechanisms considered in this paper can, in principle, generate concurrent VI VI choice proportions that are found in the literature. This is not the case for data that describing the molecular structure of choice under concurrent VI VI schedules of reinforcement.

Whereas matching is derived from thousands of responses, molecular analyses of choice look for controlling variables that act on individual responses or a handful of responses. Such analyses have indeed found regularities in responding that is suggestive of molecular control.

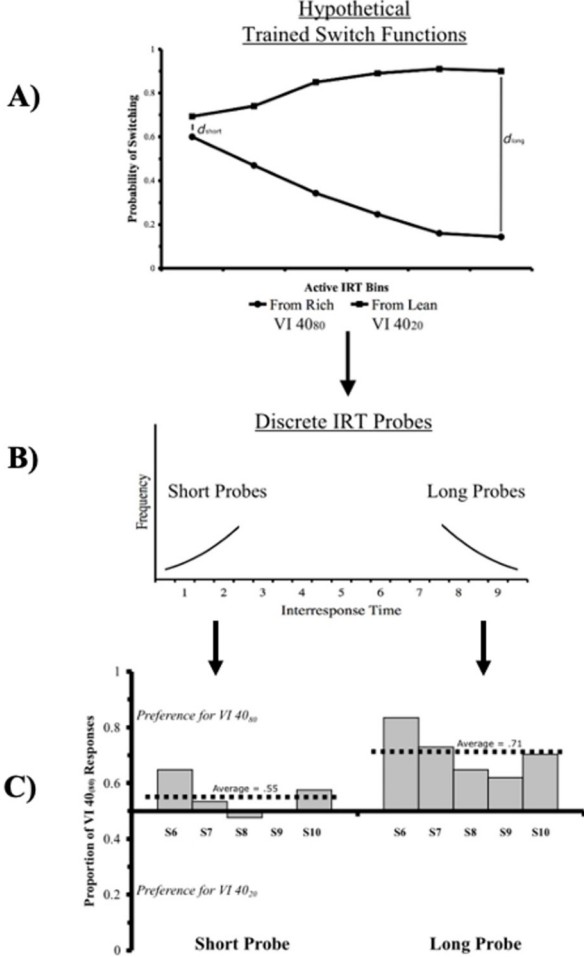

**Fig 6. Misak & Cleaveland's support of active time functions.** As noted in the text after responding to a relatively rich VI schedule, pigeons have been found to be more likely to switch after shorter than longer active IRTs. Conversely, after responding to a relatively lean VI schedule, pigeons usually show a high probability of switching across all active IRTs. A) In other words, at short active IRTs the difference between switching probabilities ($d_{short}$) for the relatively lean and rich schedules would be small in comparison to switch probabilities after long IRTs ($d_{long}$). If switch probabilities determine overall preference at suggested by ATM, then we would predict that pigeons would show more extreme schedule preferences after long active IRTs than after short active IRTs. B) Misak & Cleaveland found just such a result by utilizing probe trials in which a restricted portion of IRTs are allowed, and pigeons were given a choice between stimuli previously paired with a relatively rich VI 40-s stimulus ($VI40_{80}$) and a relatively lean VI 40-s schedule ($VI40_{20}$). C) The data revealed that during probes with short active times, birds trended towards indifference (avg = 0.55). However, probes with long active times produced a strong preference for the $VI40_{80}$ stimulus (avg = 0.71). Such data strongly supports the hypothesis that choice proportions during concurrent VI VI schedules are a function of active time functions.

For example, the most frequent pattern observed with pigeons under concurrent VI VI schedules is the pattern given by the ratio of the schedule values–a response pattern that when iterated yields the matching law [55, 56]. VIs of equal value produce more alternations than any other choice pattern. VIs with a 2:1 ratio (i.e., one VI schedule has an average programmed rate that is twice the alternative) tend to produce choice sequences dominated by two pecks at the rich schedule followed by a single peck at the poor schedule.

One way to account for these sequential dependencies is to note that they track reinforcement probabilities on a moment-to-moment basis, a type of control termed molecular maximizing. The response structures predicted by momentary maximizing may be graphically

represented if we define an indifference line at which the probabilities of reinforcement for the two choices are equal [69]. Referring to Eq 2 and setting $P_1$ equal to $P_2$, yields

$$t_1 = t_2 \left( \frac{\lambda_2}{\lambda_1} \right) \qquad (10)$$

where $t_i$ is the time since the last response to Choice 1 or Choice 2, and $\lambda_i$ is the programmed reinforcement rate for Choices 1 or 2. Note that the slope of the indifference line in the resultant *clock space* is given by the ratio of the programmed schedule values, $\lambda_2/\lambda_1$. If $t_1$ is greater than $t_2$, multiplied by this ratio, then the probability of reward for Choice 1 will be greater than the probability of reward for Choice 2.

Graphically, a clock space represents every choice that a subject emits in terms of its temporal distance from the most recent choices to each of the two concurrent VI VI schedules. These temporal distances correspond to the active and background IRTs shown in Fig 1. For example, in Fig 7 if a subject responds at a fixed rate to the VI 60-s schedule this would produce a single *x*-axis value, while the *y*-axis would increment with each choice. In other words, perseveration at a schedule produces a stream of points away from the axis that represents the IRT for that schedule. Given Eq 10, in a clock space the choice represented by the nearest axis, without crossing the indifference line, has the higher probability of reinforcement. Thus, in

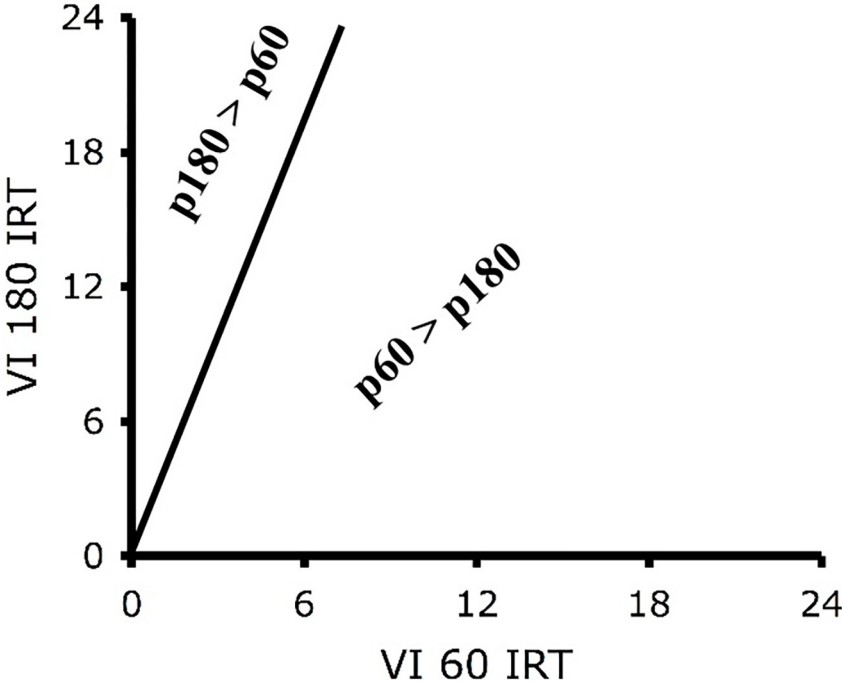

**Fig 7. Clockspace for concurrent VI VI.** Shown is a clock space for a concurrent VI 60-s VI 180-s schedule. Eq 10 provides an indifference line that indicates all points in which the reinforcement probabilities are equal at the two schedules. A point in a clock space indicates a single choice defined in terms of its temporal distance (i.e., IRTs) from the most recent selection of each of the concurrent schedules. These IRIs correspond to the active and background ITs illustrated in Fig 1.

Fig 7, the *x*-axis schedule (i.e., VI 60-s) will have the higher reinforcement probability at all points below the indifference line, while the *y*-axis (i.e., VI 180-s) schedule will have the higher reinforcement probability at all points to the left indifference line.

Momentary maximizing, as can be seen via Fig 7, predicts that if an animal responds at a constant rate, then a single response sequence will emerge that is equal to $\lambda_2/\lambda_1$. These were in fact the most frequent response sequences reported by Shimp [55] and Silberberg et al. [56]. However, an animal need not respond at a constant rate, and in such circumstances momentary maximizing does not predict a regular molecular response pattern. In order to determine whether pigeons that do not respond at a regular rate are nonetheless adhering to a momentary maximizing mechanism, Hinson & Staddon [22] developed a statistic, *M*, that captures the degree to which a pigeon's choice behavior tracks momentary reinforcement probabilities (*p*). *M* is in essence the accumulated proportion of reinforcement probability differences obtained at each choice in a session. The statistic is given by

$$M = \frac{\sum |(p_i - p_j)|_{corr}}{\sum |(p_i - p_j)|_{corr} + \sum |(p_i - p_j)|_{inc}} \tag{11}$$

where $|p_i-p_j|_{corr}$ = Choice *i* when $p_i > p_j$ or Choice *j* when $p_j > p_i$, while $|p_i-p_j|_{inc}$ if Choice *i* when $p_j > p_i$ or Choice *j* when $p_i > p_j$. If a subject's behavior tracks momentary reinforcement probabilities, *M* will approach 1.0. Behavior that is indifferent to momentary reinforcement probabilities would produce *M* values of 0.5, while sub-optimal behavior would produce *M* values less than 0.5.

Hinson & Staddon [22, 23] found that pigeons in free-operant concurrent VI VI situations tended to produce *M*-values between 0.7 and 0.9 late in training. These findings were replicated by Cleaveland [24]. Fig 8, for example, shows the average *M* values for Cleaveland's five subjects under a free-operant concurrent VI 60-s VI 180-s schedule. As can be seen there is a clear trend of increasing values. However, Cleaveland also found that pigeons in discrete-trial

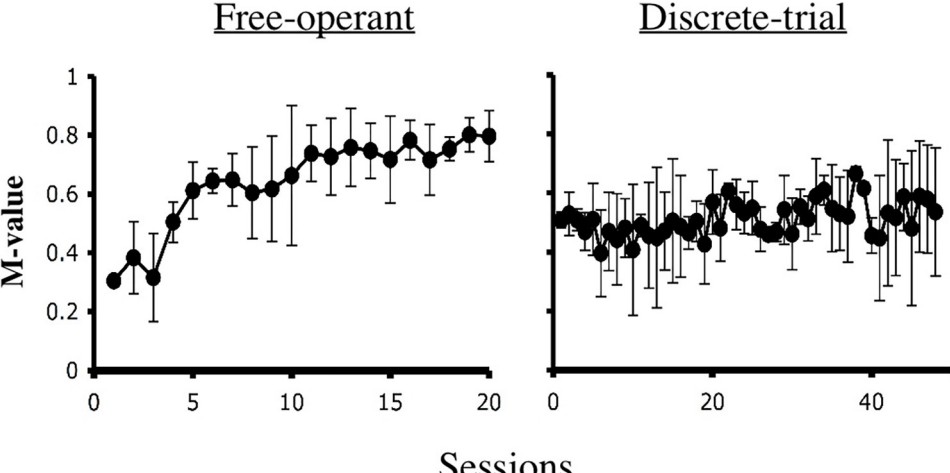

**Fig 8. Group M-values.** The figure shows the average M-values for five birds run in a free-operant procedure and for six birds run in a discrete trial procedure [24]. In both cases training involved a VI 60-s VI 180-s of reinforcement.

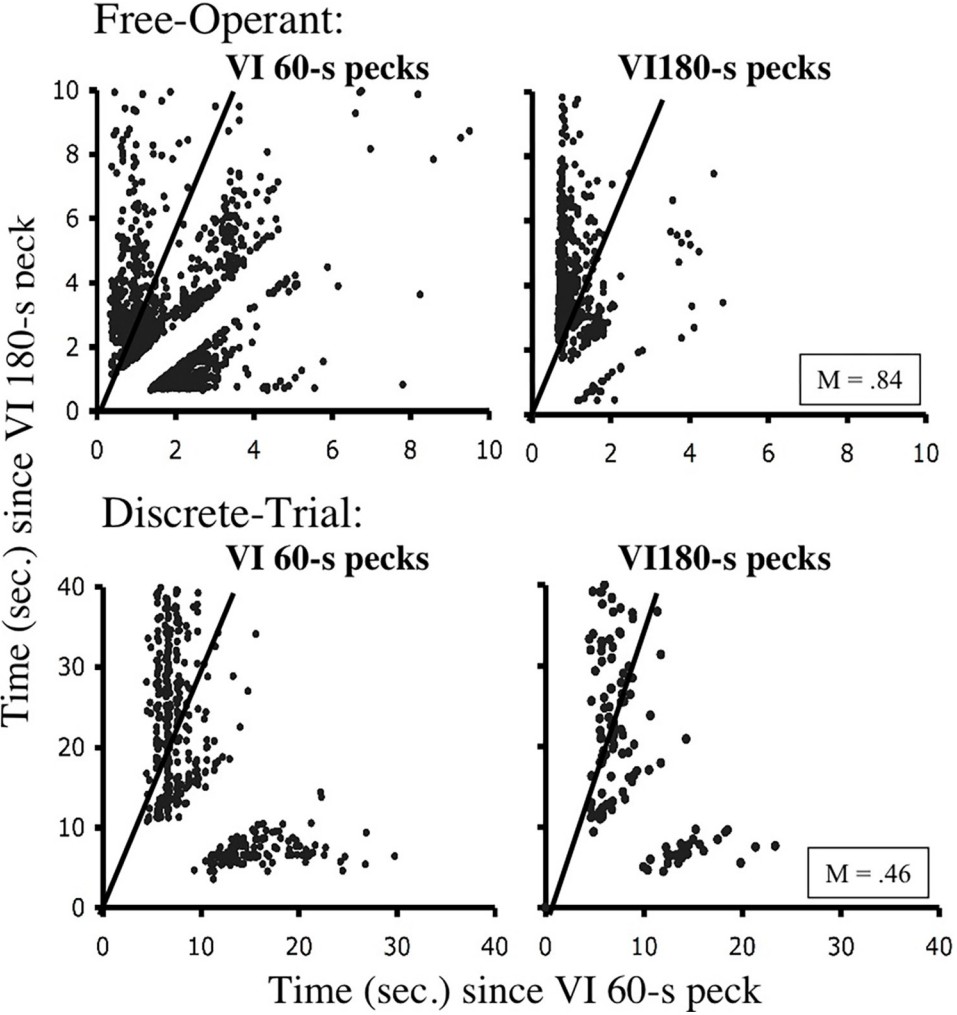

**Fig 9. Representative clock spaces from real data.** The top panel provides a clock space created by a bird experiencing a free-operant concurrent VI 60-s VI 180-s schedule of reinforcement. The lower panel provides a clock space for a bird experiencing equivalent schedules in a discrete-trial procedure [24]. Every point indicates a single peck, and the two plots for each bird separate these pecks into those made to the VI 60-s versus VI 180-s schedules. Recall that a subject tracking momentary reinforcement probabilities should produce points below the indifference line in the left-hand plot and above the indifference line in the right-hand plot. The given M-value of .84 suggests this was the general pattern for the free-operant subject. Conversely an M-value of .46 is more suggestive of behavior that is indifferent to momentary reinforcement probabilities.

procedures tended to produce *M* values of approximately 0.50 as is also illustrated in Fig 8. *M* values, though, only summarize the degree to which reinforcement probabilities might be driving choice behavior. They say nothing as to the actual patterns of responding. The latter, though, are evident when one examines clock spaces.

Fig 9 provides representative clock spaces for a subject trained in a free-operant procedure versus a subject trained in a discrete-trial procedure [24]. For clarity the clock spaces are separated into those representing pecks made to the VI 60-s and those made to the VI 180-s schedule. Cleaveland found no differences in the pattern of responding made to the lean, VI 180-s schedule. Essentially, these clock spaces indicate that subjects tended to produce a single response to the relatively lean schedule. Where the behavior generated by the two procedures differs is with regards to preservation at the rich, VI 60-s schedule. Perseveration is indicated

**Table 2. Probability maximizing for four models of VI-VI choice.**

| Model | | Free Operant | | Discrete-Trial | |
|---|---|---|---|---|---|
| | | M-Value | | M-Value | |
| | ** | VI20-VI60 | VI60-VI180 | VI20-VI60 | VI60-VI180 |
| Melioration | 15 | 0.26 | 0.31 | 0.24 | 0.31 |
| | 30 | 0.19 | 0.21 | 0.21 | 0.21 |
| | 60 | 0.2 | 0.17 | 0.2 | 0.17 |
| Momentary Max. | | 1.00 | 1.00 | 1.00 | 1.00 |
| SET* | | 0.45 | 0.46 | 0.52 | 0.45 |
| Active Time | 1/s | 0.61 | 0.72 | 0.51 | 0.53 |
| | 2/s | 0.71 | 0.72 | 0.51 | 0.54 |
| | | | | | |

Provides momentary maximization approximations via the M-metric described in the text for model simulations. For melioration three windows were utilized over which local rates were calculated. For ATM two response rates were simulated.

in Fig 9 by a vertical stream of points in the VI 60-s clock space. For the discrete-trial procedure, this stream continues well above the indifference line. Points at such a distance correspond to large probability differences that factor into the denominator of the *M* calculation. In contrast, the subject experiencing the free-operant procedure produced much shorter response runs to the VI 60-s schedule, and hence, "mistakes" weighted the denominator of the *M* calculation to a lesser degree.

Of the choice mechanisms considered in this paper, only ATM accurately fits the molecular structure of responding under free-operant and discrete-trial concurrent VI VI schedules. Table 2 shows that across the simulated schedule values, ATM produces *M* values between 0.61 and 0.71 for the free-operant procedure, and between .51 and .54 for the discrete-trial procedure. Although the values for free-operant procedure appear low, it should be remembered that the simulations used in this paper applied an average active time function. Cleaveland [24] showed that when the active time functions of individual subjects are used, the resultant *M* values are extremely similar to those produced by real subjects. Finally, as can be seen in Figs 10 and 11, the clock spaces generated by ATM show response patterns that match those produced by real birds in the free-operant and discrete-trial procedures.

In contrast to ATM molecular maximizing is clearly violated by the behavior of birds under discrete-trial VI VI schedules. Regardless of procedure, Table 2 shows that simulated choices produce M values of 1.0. These *M* values translate into the clock spaces shown in Figs 10 and 11 in which responding is clustered below the indifference line for VI 60-s responses and above the indifference line for VI 180-s responses. In other words, momentary maximizing does not produce the preservation seen in Fig 9 for discrete-trial procedures.

Similarly, both SET* and melioration produce *M* values that are at odds with those reported in the literature. In a two-choice procedure, SET* allocates choices in a stochastic manner that is independent of IRTs. Therefore, it is not surprising that simulations of SET*, in every instance, produce *M* values close to .5. These values translate into a free-operant clock space in which points are not correlated with reinforcement probabilities. Interestingly, though, the pattern of responses shown by a SET* mechanism in the discrete-trial simulation is not that different from the pattern shown by actual birds. Melioration, however, fails to produce any aspect of the molecular structure seen in real animals. With the parameters used here, melioration produced *M* values ranging from .19 to .26. These values are much lower than those observed in real animals in either the free-operant or discrete-trial procedure. Furthermore,

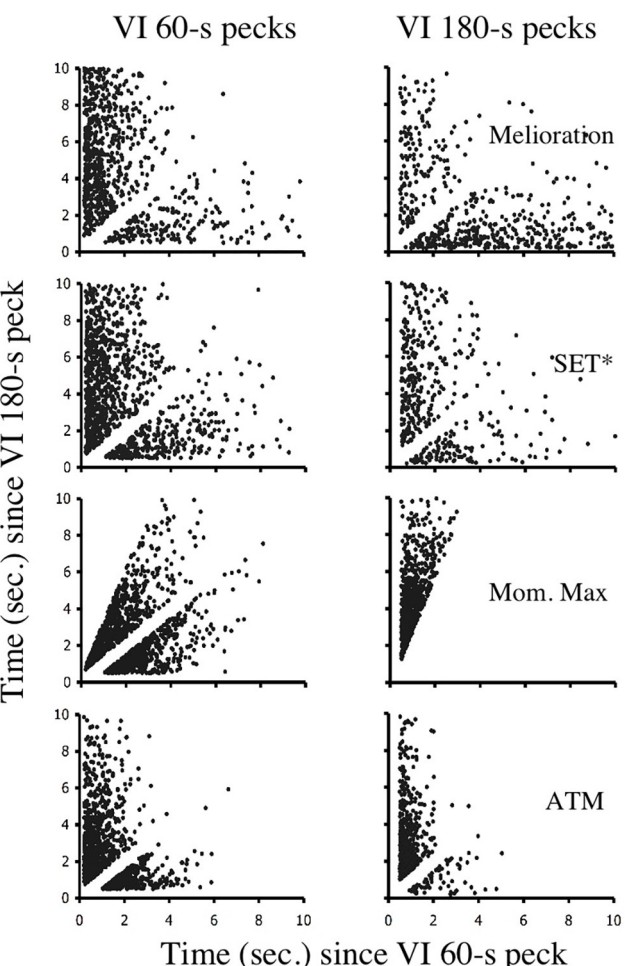

**Fig 10. Simulations of free-operant clock space data.** A free-operant concurrent VI 60-s VI 180-s schedule was simulated using four choice models: melioration, SET*, momentary maximizing, and ATM. The resultant clock spaces are given here for the last 2,000 choices of each simulation.

the clock spaces produced by melioration indicate that the mechanism generates runs of responding at both VI schedules, rather than at only the relatively rich schedule. The reason for this finding relates to the windows used in order to calculate local rates of reinforcement. Relative to the window size, reinforcers were rare events. Local rates, therefore, tended to change rather abruptly and by relatively large amounts.

## Data set 3: Run-length and choice

The response sequencing summarized by $M$ values and clock spaces is one type of molecular choice data. Another involves the probability of switching away from an alternative given a number of pecks at that choice. This latter variable is often referred to as run-length. It has been well documented that switching probabilities are independent of run-length in pigeons that are experiencing concurrent VI VI schedules [12, 24, 25, 46, 56]. That is, the probability of switching out of one schedule in a concurrent VI VI experiment appears to be independent of the number of pecks made to that schedule. Fig 12 illustrates this result with real birds experiencing a concurrent VI 60-s VI 180-s schedule [24]. The plots show the probability of a switch given the amount of time a bird has spent responding to a single alternative. In both

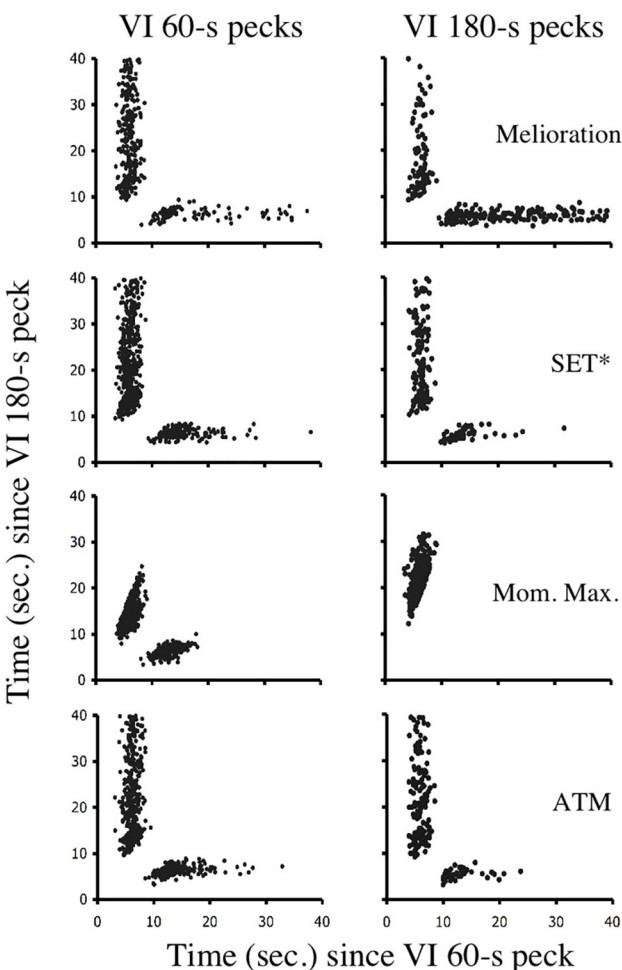

**Fig 11. Simulations of discrete-trial clock space data.** A discrete-trial concurrent VI 60-s VI 180-s schedule was simulated using four choice models: melioration, SET*, momentary maximizing, and ATM. The resultant clock spaces are given here for the last 2,000 choices of each simulation.

discrete-trial and free-operant procedures the plots are flat. For instance, in the free-operant case, birds are just as likely to switch out a rich schedule after responding to it for 1 second as they are after responding to it for 6 seconds.

Momentary maximizing does not predict the run-length data of real birds for the following reason. As Fig 1 illustrates, when a bird responds at one schedule the reinforcement probability of the background schedule grows. In other words, perseveration on one schedule is equivalent to increases in the background IRT. Momentary maximizing predicts that as the background IRT increases, the animal should become more likely to switch out of the active schedule. Allowing for errors by the animal in mapping IRTs to reinforcer probabilities, one would not expect an immediate switch. However, momentary maximizing certainly predicts a positive correlation between background IRTs and switch probabilities. Fig 13 provides simulated data for each of the four choice mechanisms considered in this paper, and as can be seen, a momentary maximizing mechanism does produce a positive correlation between runlengths and switching. This is not born out by real subjects.

In contrast to momentary maximizing, SET*, ATM and melioration all predict flat switch probabilities regardless of runlength, albeit for different reasons. SET*'s predictions are the

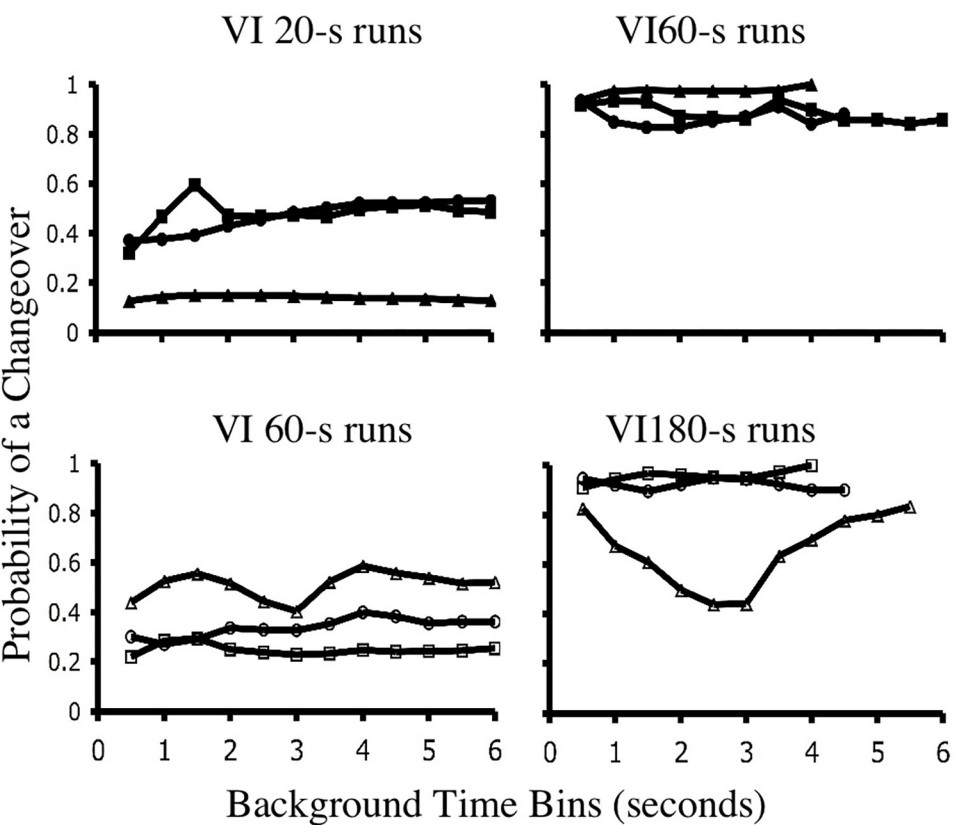

**Fig 12. Switch probability as a function of perseveration.** This figure shows the probability of switching out of a schedule given increasing background IRTs. Regardless of the time spent responding at a schedule, the likelihood of switching out that schedule remained, on average, relatively flat. The plots shown here are drawn from birds experiencing free-operant concurrent schedules of reinforcement [24]. Left-Right plots provide the trained pair of reinforcement schedules.

easiest to grasp, because it uses a stochastic decision rule whose probabilities are given by the ratio of the molar reinforcement rates at each schedule. Just as the probability of a heads on a coin toss is independent of previous coin tosses, so too are choices within SET*.

The reason why melioration produces flat switch probabilities given response runs of different lengths is slightly more complicated, as the relation between VI richness and window size is relevant. In general, though, since melioration's decision rule is based on a comparison between two averages (i.e., local rates) one would not expect there to be a correlation between runlengths and switch probabilities. The simulated run-lengths used to create Fig 13 came from a simulation that used a window size of 60 pecks. Smaller window sizes, assuming no change in VI schedules, tended to move the function associated with the VI 60 schedule (shown in Fig 13) up towards an asymptote of 0.5. This is intuitively obvious because with a window size of 1, switch probabilities depend only on baseline switching probabilities, which were set at 0.5 in the simulations.

Finally, ATM, like SET* and melioration, also produces flat switch probabilities regardless of runlength. In fact ATM makes the rather strong claim that the flat switch probabilities shown by pigeons under concurrent VI VI schedules are a sampling artifact. What matters is not the runlength but the IRT following the last peck of the run. VI schedules tend to produce steady rates of responding. Thus, the IRT after the last peck of runs with a length of 3 will, on

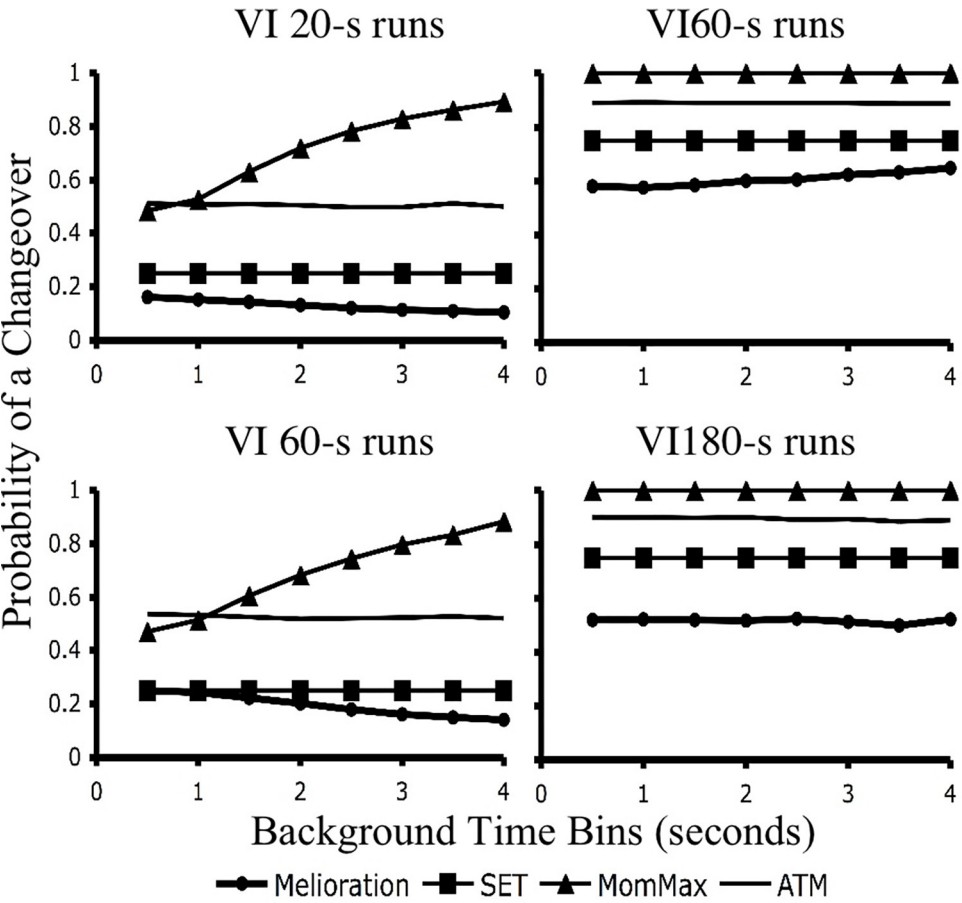

**Fig 13. Simulations of perservation.** A free-operant concurrent VI 60-s VI 180-s schedule was simulated using four choice models: melioration, SET*, momentary maximizing, and ATM. The resultant conditional probabilities of switching given runlength are given for the last 2,000 choices of each simulation. Only momentary maximizing deviated from the pattern found in real birds (Fig 12).

average, be the same as the IRT after runs with a length of 20. In other words, ATM asserts that, despite different runlengths, one is sampling the same average IRT.

### Data set 4: Multiple concurrent VI VI schedules

The final data set considered here comes from studies utilizing multiple concurrent VI VI schedules. In such studies, pairs of VI schedules are trained within a single session. For example, a subject might experience periods of a VI 20-s VI 60-s schedule intermixed with periods of a VI 60-s VI 180-s schedule. In a typical experiment the stimuli associated with each reinforcement schedule are then paired in novel combinations in order to highlight comparative preferences and, therefore, the associative processes that might be at work during training. In fact, melioration, momentary maximizing, SET* and ATM make quite specific and unique predictions for novel schedule pairings. One set of predictions was presented earlier in the paper when discussing the ability of these models to account for proportion matching (see Fig 6). Here we will consider additional data.

Fig 14 provides the predictions of melioration, SET*, momentary maximizing and ATM for pairing the two VI 60's and the two relatively rich schedules (i.e., the VI $20_{60}$ and VI $60_{180}$) from the schedule combinations just described. As can be seen, melioration predicts strong

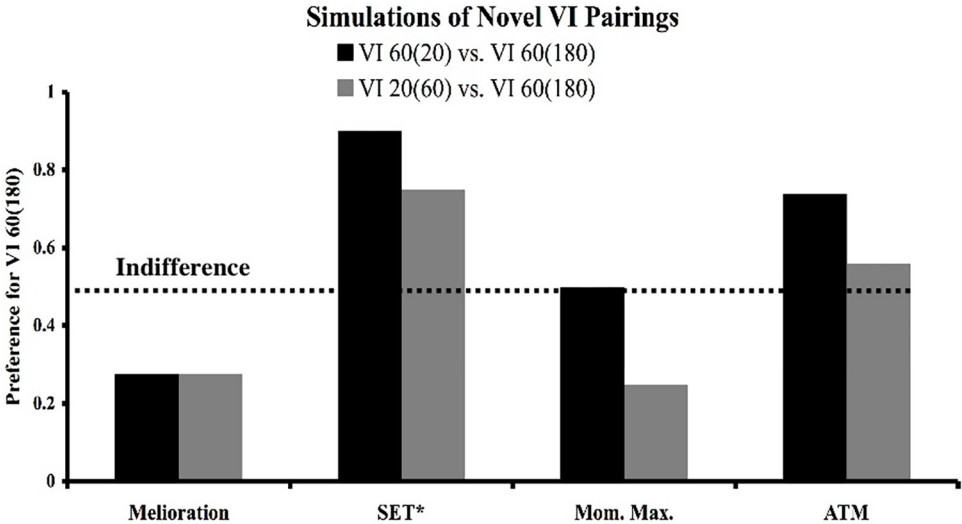

**Fig 14. Simulations of concurrent multiple VI VI probe preferences.** A powerful procedure for investigating mechanisms for concurrent VI VI choice involves training separate concurrent VI VI schedules, and then pairing the associated stimuli in novel pairings. Here, simulations for all four models were conducted for a concurrent VI VI 20-s VI 60-s schedule and a VI 60-s VI 180-s schedule. Simulated novel pairings involved the VI 6020 vs. VI 60180 and the VI 2060 vs. VI 60180 stimuli. For melioration predictions were derived by comparing the overall local rates of reinforcement accumulated during simulated training. For momentary maximizing, Eq 2 provided the predicted ratios. For ATM active time functions were drawn from Figs 4 and 5 and a response rate of 1/ was utilized. The predictions for SET* are explained in the text.

preferences for the both the VI $60_{20}$ and VI $20_{60}$ over the VI $60_{180}$. The reason for this is that overall reinforcement rates have a positive correlation with local reinforcement rates. In the simulations run here local reinforcement rates in the VI 20-s VI 60-s schedule were approximately twice as high as those produced in the VI 60-s VI 180-s schedule. Since melioration asserts that local rates drive choice, it predicts preferences for the $60_{20}$ and the VI $20_{60}$ when paired with the VI $60_{180}$. Momentary maximizing, on the other hand, predicts indifference when the two VI 60 stimuli are paired, and a preference for the VI $20_{60}$ over the VI $60_{180}$. The preferences come directly from functions given by Eq 2. The functions for the VI 60's would be identical, while the function for the VI $20_{60}$ would rise three times as fast as that for the VI$60_{180}$.

SET* and ATM make predictions for novel VI pairings that are quite different from melioration and momentary maximizing. As can be seen in Fig 14, both of these mechanisms predict a preference for the VI $60_{180}$ over the VI $60_{20}$ when their associated stimuli are paired. For SET*'s account, recall that its mechanism first assumes that subjects learn switch probabilities given by the ratio the trained, molar reinforcement rates. Therefore, a subject would switch away from a VI $60_{180}$ stimulus with p = .25 and from a VI $60_{20}$ stimulus with p = .75. This in itself would generate a 3:1 preference for the VI $60_{180}$ stimulus. However, recall that SET* also assumes that probabilities are sampled at a rate proportional to the overall rate of reinforcement for each trained pair of concurrent schedules. In this case, a VI 20-s VI 60-s schedule has an overall rate of reinforcement that is three times that of a VI 60-s VI 180-s schedule. In a novel pairing of the VI 60's this would serve to bias the preference even further for the VI $60_{180}$ stimulus, producing a preference of 9:1. ATM's prediction, on the other hand, is derived from the functions show in Figs 3 and 4. In these figures, rich-schedule switch functions are always lower than lean-schedule functions. Therefore, in the novel pairing of VI 60's ATM predicts a preference for the VI $60_{180}$ stimulus because it corresponds to a rich active time function. The

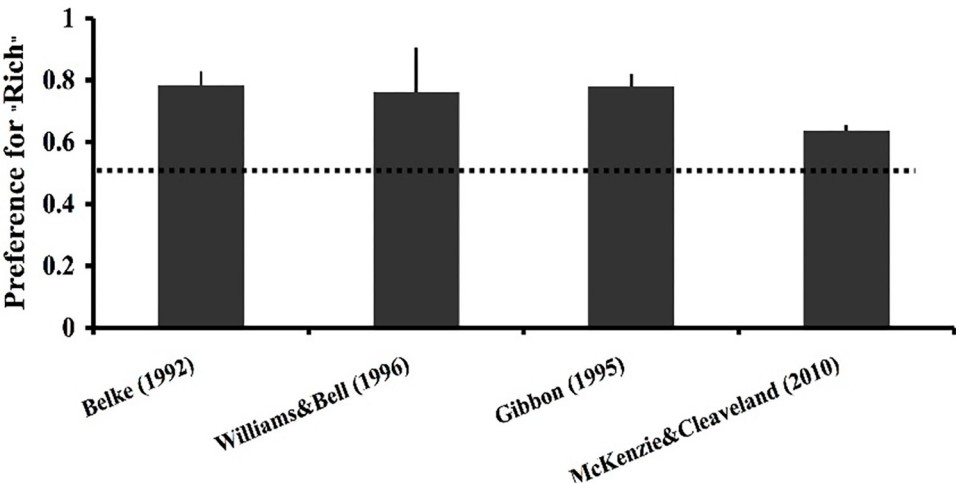

**Fig 15. Experimental replications of Belke 1992.** The results of four published studies that trained two pairs of concurrent VI VI schedules with 2:1 ratios of overall reinforcement rates. For Belke [26], Gibbon [28], and Williams & Bell [30] the schedules were a VI 20-s VI 40-s and a VI 40-s VI 80-s. For McKenzie & Cleaveland [29] the schedules were a VI 30-s VI 60-s and a VI 60-s VI 120-s. Probes involved novel pairings of the equivalent VI schedules. In all cases a preference was found for the relatively rich, though equivalent, VI schedule. Of the models considered in this paper, only SET* and ATM predict these findings.

degree of preference, though, would depend on response rates and the actual differences between the two functions at different active IRTs. The functions used here produced a 3:1 preference for the VI $60_{180}$.

Finally, whereas SET* and ATM both predict preferences in favor of the VI $60_{180}$ when the two VI 60's are paired, they make different predictions when the VI $20_{60}$ and the VI $60_{180}$ are paired. SET* make the strong and counter-intuitive prediction of 3:1 preference for the VI $60_{180}$. The reason for this is that the switch probabilities associated with each stimulus would presumably be the same (p = .25), yet the probability would be sampled three times more slowly for the VI $60_{180}$ stimulus. In contrast ATM predicts indifference between the VI $60_{180}$ ad the VI $20_{60}$ for the reason that both stimuli would presumably be associated with typical, rich active time functions.

The predictions just described have in fact been tested for each of the four models considered in this paper. The first prediction concerning the novel pairing of the two VI 60's is captured in an experiment performed by Belke [26]. In Belke's experiment, pigeons were trained under multiple concurrent VI 20-s VI 40-s and VI 40-s VI 80-s schedules of reinforcement. In probe tests, the stimuli associated with the two VI 40-s schedules were paired. Fig 15 provides the results of his experiment, which was that his birds showed a strong, 4:1 preference for the VI $40_{80}$ stimulus. Further, several labs have performed experiments similar to Belke's, and in every case a preference has been found for the stimulus trained with a relatively rich VI schedule (28–30, 68).

Belke's results, and its replications, support SET* and ATM while contradicting the predictions of melioration and momentary maximizing. Experiments seeking to distinguishing between SET* and ATM, though, have thus far been inconclusive. On the one hand, Gibbon [28] conducted a replication of Belke's experiment and paired the VI $20_{40}$ and the VI $40_{80}$ stimuli in probes. He reported a 2:1 preference in favor of the VI $40_{80}$ stimulus. McKenzie &

**Fig 16. Taken from Brown & Cleaveland [27].** All birds were first trained using multiple concurrent VI 30-s VI 30-s and VI 60-s VI 60-s schedules of reinforcement. The figure provides the preference for a VI 60-s stimulus in probe pairings that with a stimulus trained with either a VI 30-s or a VI 60-s stimulus. The dashed horizontal line indicates indifference. The simulations of SET* and ATM are described in the original article.

Cleaveland [29], though, trained pigeons under a similar arrangement–multiple concurrent VI 30-s VI 60-s and VI 60-s VI 120-s schedules—and reported indifference when the VI $30_{60}$ and VI $60_{120}$ stimuli were paired. Finally, an experiment by Brown & Cleaveland [27] also supported ATM while contradicting the predictions of SET*. In their study Brown & Cleaveland trained pigeons in multiple concurrent VI 30-s VI 30-s and VI 60-s VI 60-s schedules. They then probed their subjects with pairings of the VI 30 and VI 60 stimuli. Fig 16 provides the results of the experiment. As can be seen on average the birds showed indifference in the probe pairings. In addition ATM fit the data of individual subjects much more closely than did SET*, and as described earlier in the paper (Fig 6), ATM predicts the choice outcomes that are seen when birds are allowed to respond at different rates.

## Conclusion

This paper has shown how a relatively simple model, the active time model (ATM) can account for a range of data not previously brought under a single explanatory framework. Further, the paper showed how the assumptions underpinning ATM–IRT control of choice, relativistic response units (i.e., "switch" vs. "stay"), and Markov state spaces—are already established in the literature. Table 3 summarizes the fits produced by ATM, melioration, momentary maximizing, a version of scalar expectancy theory, SET*, to the four data sets considered in this paper.

All four models tested in this paper produced data in line with the matching law, with a tendency towards undermatching. While only ATM and momentary maximizing accurately captured the molecular structure of responding in a free-operant procedure, only ATM and SET* could account for such structure in a discrete-trial procedure. With regards to a sensitivity to background IRTs, melioration, SET* and ATM predicted the flat switch functions that have repeatedly been reported in the literature. Finally, only SET* and ATM predict the novel pairings of stimuli trained under multiple concurrent VI VI schedules, and there is reason to believe that in this data set, ATM produces the better fit. In summary, then, only ATM provided a qualitative fit for all four data sets.

**Table 3. Four VI VI choice mechanisms and their fit of four data sets.**

| | Melioration | Momentary Max. | SET* | Active Time |
|---|---|---|---|---|
| **Controlling Stimulus** | **Local Reinf rate** | **Reinforcement probabilities** | **Memory for Inter-Reinf. Times** | **Most recent, single IRT** |
| **Data Set 1** | | | | |
| Generate matching? | Yes | Yes | Yes | Yes |
| **Data Set 2** | | | | |
| Fit IRT Structure? | No | Yes | No | Yes |
| **Data Set 3** | | | | |
| Fit runlength data? | Yes | No | Yes | Yes |
| **Data Set 4** | | | | |
| Fit Belke (1992) | No | No | Yes | Yes |
| & | | | | |
| Brown & Cleaveland | No | No | No | Yes |
| (2009) | | | | |

A summary of whether the given model was found to accurately fit each of the four data sets considered in the paper. The controlling stimulus of each model is provided.

Despite ATM's obvious strengths it is not without weaknesses. Currently, the origins of the active time switch functions (Figs 3 and 4) remain unknown. Therefore, there is no principled means of predicting the model's boundaries. Will a changeover delay affect these functions? Will differential reward amounts affect these schedules? These are questions which currently must be empirically answered. However, in defense of ATM most steady state models of choice do not specify the dynamic, moment-moment mechanisms that generate their steady-state assumptions. Of the models considered in this paper, only scalar expectancy theory is truly a dynamic model. Momentary maximization does not specify how an organism learns to associate IRTs with reinforcement probabilities (just that they do), and melioration does not specify its window over which local reinforcement rates are calculated.

Relative parsimony, too, might be held up as a criticism of ATM. ATM assumes that each discriminative stimulus is associated with a function that relates active time to switch probabilities. This might appear to give ATM more latitude than melioration, momentary maximizing and SET*. Such a perspective is misleading. For example, momentary maximizing assumes that an organism also has a distinct, VI-specific switch function associated with each discriminative stimulus in a concurrent VI VI experiment. One function relates active IRTs to reinforcement probabilities, while another relates background IRTs to reinforcement probabilities. These are then somehow compared in order to produce a schedule response. Such "parameters" are somewhat hidden by the intuitively simple, "choose the higher probability of reinforcement." In addition, the functions proposed by momentary maximization simply do not have empirical support. As Data Set #3 makes clear, the typical procedures used in concurrent VI VI experiments have found no relationship between background IRTs and switch probabilities. That such a relation exists, though, is a core assumption of momentary maximization. Similar criticisms can be leveled for melioration and SET*. Melioration does not specify the window over which local rates are calculated. SET*, too, assumes unique interreinforcement functions for each discriminative stimulus as well as functions for sampling these distributions (and additional parameters not utilized in this paper). Taken together, then, ATM is at least as parsimonious as the models considered in this paper, with the addition of having each of its core assumptions grounded in empirical findings.

However, as the Introduction of this paper hopefully made clear, it is unlikely that ATM is *the* model of concurrent VI VI behavior. First, the matching law can be produced by all four of

the mechanisms described in this paper. Second, there is also evidence that experimental contingencies can bring the behavior of pigeons under the control of IRTs [e.g., 33, 34], inter-reinforcement intervals [e.g., 70], and local rates of reinforcement [e.g., 8, 9]. Given this data, it is reasonable to assume that, with the appropriate environmental configuration, momentary maximizing, SET* and melioration will prove to be the operative mechanism for a range of procedures.

Operant choice, then, is most likely multiply determined at the molar, local and / or molecular level, and the task of the psychologist is to elucidate the mechanisms that produce behavior and the local environments under which those mechanisms are operative for any given organism. From this perspective whether one controlling variable is more fundamental than another, e.g., molar reinforcement rates vs. optimality principles, is beside the point. What is required is a framework that recognizes the selection / assignment-of-credit pressures imposed on the organism [e.g., 71, 72], and the continued exploration of the types of choice mechanisms available to an organism.

## Author Contributions

**Conceptualization:** J. Mark Cleaveland.

**Data curation:** J. Mark Cleaveland.

**Formal analysis:** J. Mark Cleaveland.

**Methodology:** J. Mark Cleaveland.

**Project administration:** J. Mark Cleaveland.

**Software:** J. Mark Cleaveland.

**Validation:** J. Mark Cleaveland.

**Visualization:** J. Mark Cleaveland.

**Writing – original draft:** J. Mark Cleaveland.

**Writing – review & editing:** J. Mark Cleaveland.

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
