## [Decision Letter · Decision Letter 0]

7 Nov 2023

PONE-D-23-23962The Active Time model of concurrent choicePLOS ONE

Dear Dr. Cleaveland,

Thank you for submitting your manuscript to PLOS ONE. After careful consideration, we feel that it has merit but does not fully meet PLOS ONE’s publication criteria as it currently stands. Therefore, we invite you to submit a revised version of the manuscript that addresses the points raised during the review process.

We look forward to receiving your revised manuscript.

Kind regards,

Peter G. Roma, Ph.D.

Academic Editor

PLOS ONE

Reviewers' comments:

Reviewer's Responses to Questions

**Comments to the Author**

1. Is the manuscript technically sound, and do the data support the conclusions?

Reviewer #1: Partly

Reviewer #2: Yes

2. Has the statistical analysis been performed appropriately and rigorously? 

Reviewer #1: I Don't Know

Reviewer #2: N/A

3. Have the authors made all data underlying the findings in their manuscript fully available?

Reviewer #1: Yes

Reviewer #2: Yes

4. Is the manuscript presented in an intelligible fashion and written in standard English?

Reviewer #1: No

Reviewer #2: Yes

5. Review Comments to the Author

Reviewer #1: PONE-D-23962

This article satisfies PLOS’s criteria for publication. Those are fairly liberal criteria; the ms would not pass muster in some other journals. So my vote is a qualified yes, but with the reservations noted below.

I largely agree with the author’s perspective on factors that drive choice in these situations. But, alas, there are a lot of trees blocking the readers’ view of the forest. This article would be more appropriate in a specialty journal. Finally, it is not clear to my eyes how the author proofs out the claim that his is a good model of the phenomena that he encompasses. I believe he eschews a tough comparison, with distinctive criteria for success, with the allusion of a “concept space”, and multi-determination. All of which is likely; but does not help winnow good from bad models; his from others.

Your checklist:

1. “Claims”. The author claims that his ATM model can account for the major results in the choice between two alternative schedules of reinforcement, and can do so more parsimoniously and effectively than other standard models of choice. If supported, this would be a very worthwhile contribution to the literature. The model is synopsed in lines 136-139.

2. “Are the claims registered with those of the literature?” Yes and no. They are contingent upon projection into the “clock-space”; the divers predictions in this space for the alternative theories are not, imo, adequately laid out.

3. “Are claims supported?” Sorta. But the ms. is intricate enough that I am not sure. Table 1, for instance, is important; but I am not quite sure how to read it, or interpret it. (I could work assiduously, and it might be clear; but we don’t want indifferent readers to have to do that.)

well, now i have to qualify my vote. I did not have the momentum to plow through, to the end. Sorry. Perhaps i reflect the typical reader.

I think the argument is great, but am overwhelmed by the details.

(it is also the case that the author does not respect the use of the AIC; his ATM functions add many degrees of freedom (even thought they are empirically derived)).

Reviewer #2: The author’s active time model (ATM) is described and compared to three other models that describe choice behavior – melioration, scalar expectance theory, and momentary maximization. The ATM model is laid out well and a good tutorial of Markov chains, processes, and semi-Markov processes is provided. The author shows how the ATM model fits behavioral data better than the three other models in various ways. Overall, the paper showcases the author’s model well and should be of interest to individuals interested in understanding matching. Some points to be addressed in a revision are provided below.

1. Inputs to ATM are based on the past performance of subjects trained on the very kinds of schedules on which the various models are compared. In contrast, the other models’ inputs are based on local rates of reinforcement, momentary probabilities of reinforcement, etc., and not on actual behavior observed in previous experiments. You can simulate performance in the other models knowing only the schedule parameters and nothing of how actual subjects performed previously under similar conditions. You need to know how actual subjects behaved previously to simulate ATM performance. The author views this as a strength of ATM vs. the other models, but I take the opposite view. It is not surprising to me that ATM better fits data from real pigeons than other models when ATM is based on data from real pigeons whereas the other models are not. That the other models require only a priori assumptions, yet do so well in predicting behavior, is a strength, not a weakness, in my opinion. The author briefly mentions this difference between the models (e.g., p. 27, lines 560-562), but should elaborate on it in the discussion.

2. M-value is first presented in Table 1 on p. 25, but it is not explained until much later, on pp. 31-32. M should be described when Table 1 is presented.

3. The figures in the online submission were out of order. For example, Fig. 2 was supposed to be the graphical representation of ATM, but was actually the eighth figure presented. I eventually figured out which figure was which, but it was difficult to keep track when there are so many figures and the figure numbers don’t correspond to where they appear in the series of figures.

Minor:

--p. 6, line 119, “one” can be deleted

--p. 6, line 127, should be “associated with relatively”

--fig. 3 caption on p. 10, line 28 – should say pigeons rather than just subjects the first time around

6. PLOS authors have the option to publish the peer review history of their article (what does this mean?). If published, this will include your full peer review and any attached files.

Reviewer #1: No

Reviewer #2: No

---

## [Author Response · Author response to Decision Letter 0]

9 Mar 2024

Many thanks to the two reviewers. I respond to their comments in the "response to reviewers" document that is attached to re-submission materials.

---

## [Decision Letter · Decision Letter 1]

13 Mar 2024

The Active Time model of concurrent choice

PONE-D-23-23962R1

Dear Dr. Cleaveland,

We’re pleased to inform you that your manuscript has been judged scientifically suitable for publication and will be formally accepted for publication once it meets all outstanding technical requirements.

Kind regards,

Peter G. Roma, Ph.D.

Academic Editor

PLOS ONE

Additional Editor Comments (optional):

Reviewers' comments:

Reviewer's Responses to Questions

**Comments to the Author**

1. If the authors have adequately addressed your comments raised in a previous round of review and you feel that this manuscript is now acceptable for publication, you may indicate that here to bypass the “Comments to the Author” section, enter your conflict of interest statement in the “Confidential to Editor” section, and submit your "Accept" recommendation.

Reviewer #2: All comments have been addressed

2. Is the manuscript technically sound, and do the data support the conclusions?

Reviewer #2: Yes

3. Has the statistical analysis been performed appropriately and rigorously? 

Reviewer #2: Yes

4. Have the authors made all data underlying the findings in their manuscript fully available?

Reviewer #2: Yes

5. Is the manuscript presented in an intelligible fashion and written in standard English?

Reviewer #2: Yes

6. Review Comments to the Author

Reviewer #2: (No Response)

7. PLOS authors have the option to publish the peer review history of their article (what does this mean?). If published, this will include your full peer review and any attached files.

Reviewer #2: No

---

## [Editor Report · Acceptance letter]

22 Mar 2024

PONE-D-23-23962R1 

PLOS ONE

Dear Dr. Cleaveland, 

I'm pleased to inform you that your manuscript has been deemed suitable for publication in PLOS ONE. Congratulations! Your manuscript is now being handed over to our production team.

Kind regards, 

on behalf of

Dr. Peter G. Roma 

Academic Editor

PLOS ONE